# Unraveling Syntax:
# Language Modeling and the Substructure of Grammars

**Laura Ying Schulz** [* 1 2]   **Daniel Mitropolsky** [* 2]   **Tomaso Poggio** [2]

[*]Equal contribution
[1]ETH Zürich
[2]Massachusetts Institute of Technology

## Abstract

While language models achieve impressive results, their *learning dynamics* are far from understood. Many domains of interest – such as natural language syntax, coding languages, arithmetic – are captured by context-free grammars (CFGs). In this work, we extend prior work on neural language modeling of CFGs in a novel direction: how language modeling behaves with respect to CFG *substructure*, namely sub*grammars*. We define subgrammars, and prove a set of fundamental theorems connecting language modeling and subgrammars. We show that language modeling loss recurses linearly over its top-level subgrammars; applied recursively, the loss decomposes into losses for "irreducible" subgrammars. Under additional assumptions, and empirically, parametrized models learn subgrammars in parallel, unlike children who first master simple substructures. We find that subgrammar pretraining can improve final performance, but only for tiny models relative to the grammar, while alignment analyses show that pretraining consistently leads to internal representations that better reflect the grammar's substructure.

## 1. Introduction

Large language models (LLMs) have stunned the world by achieving sophisticated language abilities in the past few years, yet we still do not know *how* they reach such high levels of performance. Little is also known about the *process* of language acquisition. Do LLMs, for example, master simpler substructures before progressing to more complex syntax, as children do?

A major approach has been to study trained language models (for instance investigating how a trained model analyzes and uses its knowledge of a language during inference). More recently, a small but burgeoning direction has studied how neural architectures learn *Context-Free Grammars* (CFGs), a class of formal languages that broadly captures most domains of interest, such as natural languages and programming languages. The key insight is that by training models on smaller, fully controllable CFGs, training can be efficient, and one can probe for features of CFGs (specific rules, grammar size, etc). These approaches have gained us many valuable insights (as discussed in the Related Work).

However, two things have been largely unstudied until now. First, little has been shown about the *dynamics* of how models acquire language – not the static representations or logic of trained models (even in the CFG / synthetic language literature). Second, research studying CFGs has not considered that CFGs as a mathematical object have *substructure*; they decompose into "subgrammars". Indeed, in analogous research areas that study learning of abstract hypothesis classes such as polynomials, XOR functions, and modular counting, a major focus has been *studying how learning interacts with the substructure* of these function classes (e.g. the monomials that compose polynomials).

In this work, we advance the study of language modeling of CFGs by analyzing it through the subgrammar structure. Many of our results can also be viewed through the lens of studying the *dynamics* of CFG learning. In Section 4, we begin by defining several notions of *subgrammars*: *inner* subgrammars, corresponding to subtrees of CFG derivations, and *outer* subgrammars, corresponding to simplified versions of the CFG. Our definitions of subgrammars in this way are novel (though related to classic work on the algebra of CFGs), and we believe they are the right notions for studying the substructure of CFGs. The most important contribution of our work is a suite of fundamental theorems

---
[*]Equal contribution   [1]ETH Zürich   [2]Massachusetts Institute of Technology. Correspondence to: Daniel Mitropolsky <mitropol@mit.edu>.

*Proceedings of the $43^{rd}$ International Conference on Machine Learning*, Seoul, South Korea. PMLR 306, 2026. Copyright 2026 by the author(s).

showing that the loss of language modeling, or equivalently the Kullback-Leibler (KL) divergence), *obeys a recurrence over the subgrammar structure*. Under additional (strong) assumptions this implies a model learns subgrammars in parallel. Empirically, we show that small transformers trained on CFGs learn *all the subgrammars in parallel*, unlike how children acquire language.

Next, changing gears in 5 we study whether curriculum learning, by using an inductive bias and training on a subgrammar first, can improve performance: for small models, we show it can. In 5.2 we use alignment analysis to show, quite definitively, that such pre-training results in very different internal representations of the CFG: it aligns subgrammar strings, and non-subgrammar strings, respectively, resulting in internal representations that reflect the substructure of the CFG. Finally, in Section 6 we show experimentally that even models that perform well do not "know" the subgrammar structure perfectly, with the depth of recursion being the main difficulty.

Our code is publicly available at `https://github.com/laschulz/pcfg-transformer-learning`. We provide details on the transformer architecture used in all our experiments in Appendix A, as well as the definitions of all PCFGs in Appendix D. In general, we focus on scaled-down variants of nanoGPT (Karpathy, 2023), and a small set of hand-designed PCFGs that span a range of structural phenomena. In particular, we focus on factorized subgrammars, shared substructures, hierarchical recursion, and deeply nested dependencies. The setting is intentionally controlled and interpretable, allowing us to probe how models handle compositional structure while retaining exact access to the underlying generative process.

## 2. Related Work

Transformers (Vaswani et al., 2017), and language models more broadly, have been studied in two predominant research directions: improving training methods (Bubeck et al., 2023; Jaech et al., 2024; Guo et al., 2025) and probing trained models to analyze how knowledge is stored and activated during inference (Meng et al., 2022; Geva et al., 2021; Dar et al., 2022; Ferrando & Voita, 2024). Much less is known about how such models acquire language, though evidence of stage-like acquisition *reminiscent* of child language learning in GPT-2 has been reported Evanson et al. (2023).

We approach this problem via the surrogate (and theoretically significant in its own right) approach of studying the dynamics of *language models acquiring formal languages*. This complements prior work on gradiient-based learing over structured hypothesis classes (e.g. juntas, parities, modular counting) (Klivans & Kothari, 2014; Telgarsky,

2016; Abbe et al., 2024; Daniely & Malach, 2020). CFGs provide a linguistically motivated setting where recursive structure is explicit, and formal language theory offers a well-developed foundation (Cotterell et al., 2023).

Formal languages have been used to test neural models, with mixed success. RNNs and LSTMs often fail to learn sub-regular grammars despite theoretical capacity (Avcu et al., 2017), and transformers perform well on many formal languages but struggle with recursion and counter-based mechanisms (Bhattamishra et al., 2020). Other studies confirm that transformers often fail on deeply nested grammatical structures (Lampinen, 2024). Results consistently show that gradient descent, rather than model expressivity, is the limiting factor. Theoretically, self-attention has limitations for some long-range dependencies (Hahn, 2020) despite transformers' expressivity results (Pérez et al., 2021; Yun et al., 2019) (see Strobl et al., 2024 for a survery).

Closest to our work, (Cagnetta & Wyart, 2024) also study transformers trained on PCFGs to relate learning behavior to hierarchical structure in the underlying grammar; however, the main differences are that in their work (1) the PCFGs always produce *finite support* languages (i.e. no recursion) and prediction is always of the *final* token only. As a result of these differences, in striking contrast to our results demonstrating *parallel* learning of subgrammars in autoregressive language modeling, their results show learning occurring in discontinuous "stages" as the model has enough data to utilize PCFG substructure farther and farther back from the end of the subsequence. Reconciling these approaches into a single, stronger theoretical framework is left to future work. Finally, (Allen-Zhu & Li, 2023) provide a thorough analysis of how trained transformers can implement CFG-like computations and encode rewrite rules in their internal states. Our work builds on this by studying the learning *dynamics*, specifically vis-a-vis subgrammars.

## 3. Preliminaries and Definitions

### 3.1. Formal Languages

**Definition 3.1** (CFG). A Context-Free Grammar (CFG) is a tuple $G = (\Sigma, \mathcal{N}, \mathcal{S}, \mathcal{P})$ where $\Sigma$ is a finite set of terminal symbols, $\mathcal{N}$ is a finite set of non-terminal symbols, $S \in \mathcal{N}$ is the designated start symbol, and $\mathcal{P}$ is a finite set of production rules of the form $A \to \alpha$ where $A \in \mathcal{N}$ and $\alpha \in (\mathcal{N} \cup \Sigma)^*$ ($\alpha$ can be the empty string $\epsilon$).

The language $L_G \subseteq \Sigma^*$ associated with a CFG $G$ is the set of all strings of terminals derived from $S$ using rules in $\mathcal{P}$.

**Definition 3.2** (PCFG). A Probabilistic Context-Free Grammar (PCFG) is a context-free grammar $G = (\Sigma, \mathcal{N}, \mathcal{S}, \mathcal{P})$ augmented with a probability function $\mathcal{W}$ that assigns to each rule $(A \to \alpha) \in \mathcal{P}$ a probability, such that for each $A \in \mathcal{N}$, $\sum_{\{(A \to \alpha) \in \mathcal{P}\}} \mathcal{W}(A \to \alpha) = 1$.

CFGs were originally defined in the context of linguistics (Chomsky, 1956), as the vast majority of the syntax of natural languages, as well as the syntax of programming languages and mathematics, can be formulated as CFGs (Shieber, 1985; Pullum & Gazdar, 1982). Since CFGs capture languages with recursion and embedded structure, there intuitively exists a notion of a *sub*grammar. However, several subtleties crop up when attempting to define subgrammars. We propose two notions of subgrammars, each of independent interest and relevance: one is the grammar of *substrings* of CFG sentences that can be generated from a non-terminal, and the other as a subset of the CFG language generated by a subset of the *rules*. We term these *inner* and *outer* subgrammars respectively. Intuitively, inner subgrammars correspond to *subtrees* of derivations of CFGs, whereas outer subgrammars are a simplified version of the grammar. We will sometimes say *supergrammar* for a bigger grammar containing a subgrammar.

**Definition 3.3** (Inner Subgrammar). An *inner subgrammar* of a PCFG $G = (\Sigma, \mathcal{N}, \mathcal{S}, \mathcal{P}, \mathcal{W})$ is itself a PCFG $G' = (\Sigma', \mathcal{N}', \mathcal{S}', \mathcal{P}', \mathcal{W}')$ such that $\Sigma' \subseteq \Sigma, \mathcal{N}' \subseteq \mathcal{N}$, and $\mathcal{P}'$ is the set of all rules expanding non-terminals in $\mathcal{N}'$. Finally $\mathcal{W}'$ is the restriction of $\mathcal{W}$ to $\mathcal{P}'$, i.e. renormalized so that for every $A \in \mathcal{N}'$, $\sum_{\{(A \to \alpha) \in \mathcal{P}'\}} \mathcal{W}'(A \to \alpha) = 1$ .

**Definition 3.4** (Proper Subgrammar). A *proper* subgrammar is an inner subgrammar $G'$ of a CFG $G$ which which is not the whole grammar (in particular $S \notin \mathcal{N}'$).

**Definition 3.5** (Outer Subgrammar). An outer subgrammar of a PCFG $(\Sigma, \mathcal{N}, \mathcal{S}, \mathcal{P}, \mathcal{W})$ is a PCFG $G' = (\Sigma', \mathcal{N}', \mathcal{S}, \mathcal{P}', \mathcal{W}')$, with $\Sigma' \subseteq \Sigma, \mathcal{N}' \subseteq \mathcal{N}, \mathcal{P}' \subseteq \mathcal{P}$, where $\mathcal{W}'$ is the renormalized restriction of $\mathcal{W}$ to $\mathcal{P}'$. To be a valid outer subgrammar, $\mathcal{P}'$ must contain at least one rule from $\mathcal{P}$ where the left-hand side is $S$.

An outer subgrammar captures the notion of a subset of the *language* generated by a PCFG obtained by keeping a subset of expansions of some non-terminals (including $S$). Every string generated by an outer subgrammar is a valid string of the supergrammar. An outer subgrammar more closely corresponds to the notion of a "simple" version of a language, whereas inner subgrammars are the inherent *compositional* substructures of a CFG.

### 3.2. Language Modeling

In this work, all distributions are assumed to be over strings of a finite alphabet $\Sigma$, although most definitions can be extended to arbitrary domains.

**Definition 3.6** (Kullback-Leibler Divergence). Given distributions $P$ and $Q$ over $\Sigma^*$, the Kullback-Leibler (KL) Divergence of $Q$ from $P$ is

$$\mathrm{KL}(P \parallel Q) = \sum_{s \in \Sigma^*} P(s) \log \frac{P(s)}{Q(s)}$$

A *language model* $Q_\theta$ is a function family parametrized by $\theta$, such that $Q_\theta(x)$ yields a probability distribution over $x \in \Sigma^*$. In this work one can think of all $Q_\theta$ as autoregressive (though for several theoretical results this is not strictly necessary), meaning $Q_\theta(x)$ is an abstraction of a next-token prediction model, i.e. $Q_\theta(x_1, \ldots, x_n) = \Pi_{i=1}^n Q_\theta(x_i | x_1, \ldots, x_{i-1})$.

In Language Modeling, $Q_\theta$ is optimized with Maximum Likelihood Estimation:

**Definition 3.7** (Maximum Likelihood Estimation). Given a target distribution $P$, the Maximum Likelihood Estimator $Q_{\hat{\theta}}$ is parametrized by $\hat{\theta} = \arg \max_\theta \mathcal{L}(\theta)$ where

$$\mathcal{L}(\theta) = \mathbb{E}_{s \sim P} \left[ -\log Q_\theta(s) \right]$$

Practically, this is done by maximizing the combined likelihood of a set of samples, or equivalently, minimizing the sum of negative log-likelihoods; in the limit, this exactly approaches $\mathcal{L}(\theta)$.

**Definition 3.8** (Shannon Entropy). The Shannon Entropy of a probability distribution is

$$H(P) = \mathbb{E}_{s \sim P} \left[ \log P(s) \right]$$

**Proposition 3.9.** *Given a true distribution $P$ and model $Q_\theta$ parameterized by $\theta$,*

$$\mathcal{L}(\theta) = D_{\mathrm{KL}}(P \parallel Q_\theta) + H(P)$$

The proof (given in the Appendix C) is a straightforward application of the linearity of expectation. In particular, this implies that $\theta$ minimizes $\mathcal{L}(\theta)$ if and only if it minimizes $D_{\mathrm{KL}}(P \parallel Q_\theta)$.

## 4. The Fundamental Relation of Language Modeling and Subgrammars

**Theorem 4.1** (Unique decomposition). *Every (P)CFG $G$ can be uniquely decomposed into a hierarchy of its inner subgrammars.*

*This hierarchical structure can be represented as a directed acyclic graph (DAG) with self-loops, where each node is labeled by the set of non-terminals that generate the corresponding subgrammar.*

The straightforward proof recursively constructs the DAG by first identifying the "top-level" subgrammars of $G$; see Appendix C. While to our knowledge this theorem in its formulation is our own, the nodes of the DAG decomposition correspond to the "grammatical levels" of a CFG in Gruska's classical work on CFG theory (Gruska, 1971).

## 4.1. Subgrammars and Language Modeling

We now study the connection between the subgrammar structure and language modeling. Let $G = (\Sigma, \mathcal{N}, S, \mathcal{P}, \mathcal{W})$ be a PCFG that induces a distribution $P_G$ over $\Sigma^*$, and $Q_\theta$ an autoregressive language model trained to approximate $P_G$ (that is, given a sequence in $\Sigma^*$ it outputs a terminal, or EOS).

For motivation, first consider the simple case where the only expansion of $S$ is $S \to \alpha A\beta$, where $A$ is some proper subgrammar (does not generate $S$), and $\alpha, \beta \in \Sigma^*$ are strings of terminals. Then,

$$D_{\mathrm{KL}}(P_G \parallel Q) = \sum_{a \in \Sigma^*} P_G(\alpha a\beta) \log \frac{P_G(\alpha a\beta)}{Q_\theta(\alpha a\beta)}$$

$$= \sum_{a \in \Sigma^*} P_G(\alpha a\beta) \log \frac{P_G(\alpha \mid \epsilon) P_G(a \mid \alpha) P_G(\beta \mid \alpha a)}{Q_\theta(\alpha \mid \epsilon) Q_\theta(a \mid \alpha) Q_\theta(\beta \mid \alpha a)}$$

$$= \frac{\log P_G(\alpha \mid \epsilon)}{\log Q_\theta(\alpha \mid \epsilon)} + \sum_a P_A(a) \frac{\log P_A(a)}{\log Q_\theta(a \mid \alpha)} + \sum_a P_G(a \mid \alpha) \frac{\log P_G(\beta \mid \alpha a)}{\log Q_\theta(\beta \mid \alpha a)}$$

In an abuse of notation, above $P_G(\alpha|\epsilon)$ denotes the probability of a (partial) sequence that starts with $\alpha$, $P_G(a|\alpha)$ the probability of $a$ following $\alpha$, and so on. Importantly, the decomposition of $P_G$ and $Q_\theta$ follows from the *subgrammar* structure of $G$ in the case of $P_G$, and from the fact that $Q_\theta$ is *autoregressive* for the $Q_\theta$ terms. In short, *the KL-divergence evaluates to a sum of "conditioned KL-divergences" of the* subgrammar $A$, of prefix $\alpha$, and of suffix $\beta$. The latter terms can themselves be thought of as simple subgrammars; indeed, we can rewrite $G$ to include two new non-terminals that evaluate to $\alpha$ and $\beta$ respectively (with prob. 1), and we would then have a sum over three "sub-divergences".

**Definition 4.2.** Given PCFG distribution $P_G$, arbitrary distribution $Q$ over $\Sigma^*$, and top-level subgrammar $A$ of $G$, define $D_{\mathrm{KL}}(P_G \parallel Q)_A$

$$= \sum_{s \in \Sigma^*} P(s|\epsilon) P_G(A|s) \sum_{a \in \Sigma^*} D_{\mathrm{KL}}(P_G \parallel Q(\cdot|s))$$

$D_{\mathrm{KL}}(R \parallel Q)_A$ can be seen as the "restriction" of the KL-divergence to the subgrammar $A$ (by averaging over all contexts that can begin $A$). In the case of a fixed string $\alpha \in \Sigma^*$ we will write $D_{\mathrm{KL}}(P_G \parallel Q)_\alpha$ where the second sum is replaced with a single term for $\alpha$ (as one can view $\alpha$ as a subgrammar of one string). Then we have, in our previous example, $D_{\mathrm{KL}}(P_G \parallel Q) = D_{\mathrm{KL}}(P_G \parallel Q)_\alpha + D_{\mathrm{KL}}(P_G \parallel Q)_A + D_{\mathrm{KL}}(P_G \parallel Q)_\beta$.

Our first fundamental result is that this decomposition, or recurrence, holds generally. Let the *top-level* subgrammars denote the children of the root node in a CFG's subgrammar decomposition.

**Theorem 4.3** (KL loss as a recursive function over subgrammars)**.** *Let $G$ be a PCFG with top-level subgrammars $A_1, \ldots, A_k$. Let $C \subset \Sigma^*$ be the set of (fixed) substrings of terminals in expansions of $S$. Then $D_{\mathrm{KL}}(P_G \parallel Q_\theta)$*

$$= \sum_{i=1}^{k} D_{\mathrm{KL}}(P_G \parallel Q_\theta)_{A_i} + \sum_{\alpha \in C} D_{\mathrm{KL}}(P_G \parallel Q_\theta)_\alpha$$

**Corollary 4.4.** *If we rewrite $G$ as an equivalent PCFG with additional non-terminals such that $S$ maps to strings only non-terminals (corresponding to subgrammars $A_1, \ldots, A_k$); then the right sum of Theorem 4.3 can be removed:*

$$D_{\mathrm{KL}}(P_G \parallel Q_\theta) = \sum_{i=1}^{k} D_{\mathrm{KL}}(P_G \parallel Q_\theta)_{A_i}$$

The full proof of Theorem 4.3 and Corollary 4.4 are in Appendix C. Upon closer inspection, it is clear that the recursion actually applies to any *subgrammar*; that is, for subgrammar $A$ with subgrammars $B_1, \ldots, B_l$, $D_{\mathrm{KL}}(P_G \parallel Q_\theta)_A = \sum_j D_{\mathrm{KL}}(P_G \parallel Q_\theta)_{B_j}$ (indeed, we could have states Theorem 4.3 with respect to subgrammars, as $D_{\mathrm{KL}}(P_G \parallel Q_\theta) = D_{\mathrm{KL}}(P_G \parallel Q_\theta)_G$). Hence, this formula can be expanded recursively over each of the *subgrammars* $A_i$ by repeated application, resulting in a sum over all the *leaves* of the DAG decomposition of $G$ into its subgrammars; see Corollary C.1 in the Appendix for the precise statement.

Now, suppose each top-level subgrammar $A_i$ occurs with probability $p_i$ over the top-level rules that expand $S$; it is tempting to conclude that the recursive formula simplifies to $KL(P_G \parallel Q_\theta) = \sum_{i=1}^{k} p_i D_{\mathrm{KL}}(P_G \parallel Q_\theta)$ (where the KL terms are no longer restrictions, but bona-fide divergences between the distribution $P_G$ and $Q_\theta$ as a language model for $A$). However, this is true only if $Q_\theta$ is excellent and models $P_A$ identically under any context where the subgrammar $A$ can occur, which may not be the case!

**Corollary 4.5.** *Let $G$ be a PCFG where $S$ evaluates to rules with only non-terminals (correspondingly, subgrammars) $A_1, \ldots, A_k$ each of which occurs with prob. $p_i$.*

*Assume $Q_\theta$ is "context insensitive" for each grammar $A_i$: that is, for two contexts $s, s'$ for which $P_G(A_i|s) P_G(A_i|s') > 0$, $Q_\theta(A_i|s) = Q_\theta(A_i|s')$ (the restrictions of $Q_\theta$ to strings from $A_i$ given possible contexts $s$ or $s'$, are the same). Then*

$$D_{\mathrm{KL}}(P_G \parallel Q_\theta) = \sum_{i=1}^{k} p_i D_{\mathrm{KL}}(P_{A_i} \parallel Q_\theta(A_i))$$

*where $Q_\theta(A_i) = Q_\theta(A_i|s)$ for arbitrary context $s$ s.t. $P_G(A_i|s) > 0$.*

Several comments are in order about this Corollary, which simplifies the general recursive formula of Theorem 4.3

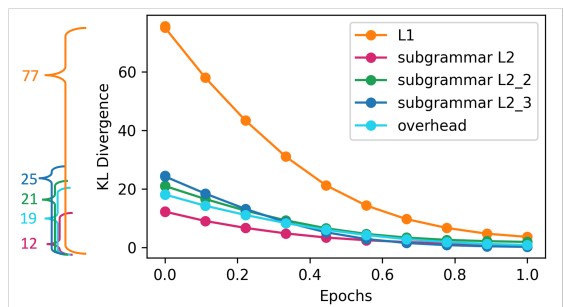

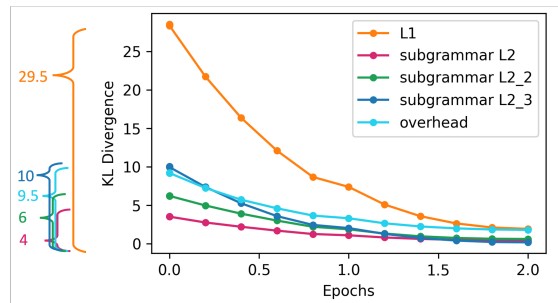

*(a)* $S \rightarrow L2\ L2\_2\ L2\_3$, grammar with inner subgrammars each occuring with 100% probability.

*(b)* $S \rightarrow L2\ (0.3),\ S \rightarrow L2\_2\ (0.3), S \rightarrow L2\_3\ (0.4)$, a grammar with inner subgrammars each occurring with < 100% probability. Each $D_{\mathrm{KL}}$ term is weighted by its prob. $p$, so to verify the claim of Theorem 4.6, the reader need only check that the displayed terms sum correctly.

*Figure 1.* Experiments visualizing decomposition of KL-divergence (Theorems 4.3 and 4.5) over subgrammars, and parallel learning of subgrammars "overhead" refers to constant strings in between subgrammar roots. Full grammars defined in Appendix D.

so that the recursive terms are simple KL-divergences (not "conditioned" KL-divergences). The corollary requires the strong assumption that the model be "context insensitive" for its subgrammars, but results in a particularly elegant decomposition. When one considers a model being *trained* over time, it may or may not context insensitive for a given subgrammar at different steps; but at any point that it *is*, this fundamental recurrence must hold. While we do not present it formally out of interest of space, it is not hard to extend to approximate / statistical versions of this corollary: to the extent that $Q_\theta$ is *not* context-insensitive, the difference between the elegant decomposition and the true loss will differ to the same extent. Finally, our experiments suggest that this condition is perhaps not so strong, at least in the statistical sense: in the experiments in Figures 1a, discussed below, qualitatively similar results were obtained when we computed subgrammar divergences with varying prefixes. In Section 6, we find that for prefixes of increasing length, our small transformer models the distribution of the ensuing subgrammar identically, but *not* if the prefixes are highly *deep*; however, such strings are "rare" under the actual distribution (so one could indeed say that these models appear to be "context-insensitive statistically").

In another direction, consider that in Theorem 4.3 and its corollaries, any of the top-level subgrammars $A_i$ could have been the grammar $G$ itself (if $G$ has a self-loop). It turns out we can say even more about the KL-divergence as a function of the *degree* of "self-loopiness", or recursion.

**Theorem 4.6** (KL-divergence with expected recurrence)**.** *Let $G$ have* proper *top-level subgrammars $A_1, \ldots, A_k$, each occurring with prob. $p_k$ over rules expanding $S$, and let $Q_\theta$ be context-insensitive for subgrammars of $G$.*

*Let the* recursion $R$ *be the number of times $S$ occurs in the rule that expands $S$. Then,*

$$D_{\mathrm{KL}}(P_G \parallel Q_\theta) = \frac{\sum_{i=1}^k D_{\mathrm{KL}}(P_{A_i} \parallel Q_\theta(A_i))}{1 - \mathbb{E}[R]}$$

*If $1 - \mathbb{E}[R] < 0$, then the KL-divergence is* unbounded *if $D_{\mathrm{KL}}(P_{A_i} \parallel Q_\theta(A_i)) > 0$ for any $A_i$.*

See C for the full proof. Theorem 4.6 can be seen as the equation for the "base case" in the recursive formula for KL-divergence, since an irreducible (leaf) subgrammar evaluates only to strings of terminals and itself! This equation shows that the expected recursion in such a (sub)grammar must be less than 1 (and the closer it is to 1, the greater the "blow-up" of its divergence to a language model); indeed, if the expected recursion is 1 or greater, the PCFG sampling process that recursively expands the root symbol will in expectation never terminate. Note that a similar, but more clunky, theorem can be shown without assuming context-insensitivity to subgrammars (replacing KL-divergences with conditioned / averaged versions).

Finally, Theorem C.2 proves a similar, recursive decomposition for *outer* subgrammars; the statement and proof have been moved there for brevity.

### 4.2. Experiments and Visualizations

To visualize these theorems, we train small transformers on synthetic PCFGs with varied subgrammar structure, and plot the KL-divergence over training in Figure 1. These example plots show visually how, throughout all stages of learning, the KL divergence (loss) is the sum over the corresponding loss for each subgrammar.

To illustrate Theorem 4.6, consider a simple CFG with two rules:

$$S \rightarrow x\ (p), \quad S \rightarrow (S \text{ and } S)\ (1 - p)$$

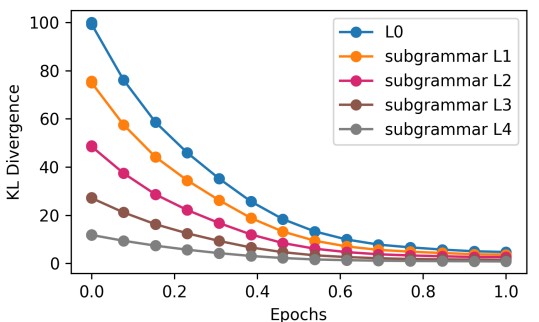
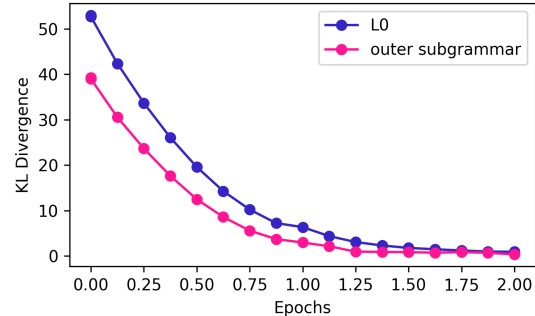

*(a)* `Deeper Recursion`: a language with an inner subgrammar DAG of depth 4.

*(b)* KL decomposition for `Outer Subgrammar Example` using most of the rules (see Theorem C.2)

*Figure 2.* Additional examples of how loss, or KL-divergence, behaves with respect to varying subgrammar structure.

The expected recursion is $\mathbb{E}[R] = 2(1 - p)$. Assuming the language model is context-insensitive, then the KL-divergence is $C/(2p - 1)$ for some constant $C$. We train a small transformer over this language with increasing $p \in (0.5, 1]$, demonstrating qualitatively the non-linear (inverse proportional) growth of KL-divergence as $p$ (the prob. of *no* recursion) approaches 0.5; see Figure 5 in Appendix E.2. Finally, we use similar experiments to visualize loss decomposition for a grammar with many nested subgrammars (i.e. the DAG is a line), and for outer subgrammars, in Figure 2.

However, an additional phenomenon jumps out from all of these plots: *the models learn all subgrammars in parallel*! One might have intuitively expected a model to first master a simpler subgrammar before progressing to the encompassing supergrammar. While the loss decomposition theorems show that nothing within the task of language learning *prevents* parallel learning of subgrammars, it is possible to construct pathological, theoretical scenarios where a model independently optimizes subgrammars in sequence. Parallel subgrammar learning is property of the training method and model architecture. We believe that our work opens a fascinating new direction of studying *when and why models learn all subgrammars in parallel*. Here, we present a simplistic but fundamental scenario in which this occurs:

**Corollary 4.7.** *(Stated informally) Suppose $Q_\theta$ is trained on PCFG $G$ with subgrammars $A_1, \ldots, A_k$ via gradient descent, and that the model and PCFG together obey a kind of "independence": for a gradient update to $\theta$ on a subgrammar $A_i$, that is $\delta = \nabla_\theta(-D_{\mathrm{KL}}(P_G \parallel Q_\theta)_{A_i})$, applying it does not hinder performance on other subgrammars. That is, for $\theta' = \theta + \delta$, $D_{\mathrm{KL}}(P_G \parallel Q_{\theta'})_{A_j} \leq D_{\mathrm{KL}}(P_G \parallel Q_\theta)_{A_j}$ for $j \neq i$ (in fact it is sufficient for this condition to hold only for $\theta$ within the path of descent).*

*Then, if trained with gradient descent, $Q_\theta$ learns all subgrammars in parallel.*

Note that this is indeed a Corollary; it would *not* always

be true if loss did not recurse linearly over subgrammars. one immediate future direction would be to study whether the small transformers and PCFGs of this paper learn subgrammars in parallel because they satisfy the independence condition of 4.7; we believe this to be the case. Despite their small size, they are likely still overparametrized with respect to the even tinier PCFGs. Future work can also aim to weaken the assumptions for parallel learning.

## 5. Subgrammars and Improving Optimization

While the previous section establishes a mathematical relationship between training loss and subgrammar structure, it is natural to consider whether the structure of CFGs could be exploited in training; e.g. is pretraining on a subgrammar helpful, and/ does it lead to a different representations? Perhaps mastering simpler components first facilitates learning of more complex structures later. This question connects to *curriculum learning* (Bengio et al., 2009; Wang et al., 2021), modular pretraining (Andreas et al., 2016; Kaiser et al., 2017), and recent work on pretraining ("pre-pretraining") on data with latent structure similar to language (e.g. music, code, formal structures) can transfer to language modeling (Papadimitriou & Jurafsky, 2020; 2023; Hu et al., 2025).

### 5.1. Robustness to Subgrammar Location

One might expect the choice of subgrammar to influence learning, given the autoregressive nature of transformers. In particular, a *prefix subgrammar*, an inner subgrammar always occurring at the beginning of sequences of G, might be easier to retain, whereas the results from pretraining on a *suffix subgrammar* or an *infix subgrammar* (appearing in the middle and disconnected from sentence endpoints) might be overwritten when training on the full grammar begins. However, our results show this is not the case: small transformers reliably retain modeling performance on *any* subgrammar, regardless of its position. This robustness is illustrated in Figure 3. As the experiments of the following

| | Two-layer Transformer | | | | Four-layer Transformer | |
|---|---|---|---|---|---|---|
| | Pretraining 10 epochs | | Pretraining 20 epochs | | Pretraining 10 epochs | |
| | Attention | MLP | Attention | MLP | Attention | MLP |
| **Full grammar sequences** | | | | | | |
| From Scratch | 0.258 | 0.535 | 0.249 | 0.535 | 0.249 | 0.469 |
| With Pretraining | 0.281 | 0.534 | 0.303 | 0.511 | 0.323 | 0.491 |
| *Percentage change (%)* | *+8.9* | *-0.2* | *+21.7* | *-4.7* | *+8.3* | *+1.0* |
| **Subgrammar sequences** | | | | | | |
| From Scratch | 0.298 | 0.561 | 0.288 | 0.558 | 0.295 | 0.513 |
| With Pretraining | 0.339 | 0.566 | 0.348 | 0.544 | 0.347 | 0.525 |
| *Percentage change (%)* | *+13.8* | *-0.1* | *+20.8* | *-2.6* | *+10.7* | *+1.9* |
| Subgrammar pretraining only | 0.288 | 0.558 | 0.288 | 0.558 | 0.295 | 0.523 |

*Table 1.* Average linear CKA similarity (0–1) across attention and MLP layers of a different, independently trained transformers when pretraining for 10 vs. 20 epochs. After pretraining, the models were trained for additional 45 epochs. The average was computed over off-diagonal seed pairs (30 seeds; 435 pairs) on the evaluation set. The most significant differences are in teal. This experiment was run using `PythonPCFG`.

section suggest, it appears that training on a subgrammar ferries the model into a distinct area of weight space in which the subgrammar is internally represented, and further optimization (on the whole language) remains in this subspace.

### 5.2. Activation-space analysis

We examine how subgrammar pretraining affects internal representations by comparing models trained from scratch to those pretrained on a subgrammar and then continued on the full grammar. In this experiment, all models are trained for the same total number of epochs and total number of sentences. Concretely, for each condition we train 30 random initializations for 2 settings: (1) the set of models is trained directly on the full grammar for $x$ epochs, (2) the set of models is pretrained for $y$, where $y < x$ epochs on the subgrammar followed by $x - y$ epochs on the full grammar. This setting was then ablated for different $x, y$. Similarity is measured with Centered Kernel Alignment (CKA) (Kornblith et al., 2019) across 30 random seeds. Independently trained models don't need to align neuron-by-neuron: the same computation can be represented after permutations or rotations of the hidden basis. CKA compares the geometry induced by activations across evaluation tokens, making it a natural measure of cross-seed representational similarity; Appendix B provides details on the method and its application in our experiments.

Much to our surprise, we also found that for smaller models, subgrammar pretraining can even help achieve a *lower final loss* (Figure 6 in Appendix E.3). This is striking because the pretrained models spend fewer epochs optimizing on the full grammar. Nevertheless, their internal representations of subgrammar strings remain more similar across seeds after full-grammar training, suggesting that these representations are not erased. More surprisingly, representations of full-

grammar strings also become more aligned across seeds. This effect diminishes as the model size and representational complexity increase (for instance, this occurs for 2-layer transformers but not 4-layers). As expected, larger models consistently reach lower losses regardless of pretraining.

CKA analysis reveals that *pretrained models exhibit higher alignment across attention layers than models trained from scratch*, both when computed over (1) subgrammar sequences (surprising because these representations are not "erased" in the second phase when optimizing for the full grammar) and (2) even more surprisingly, full-grammar sequences (on which the pre-trained models have been trained for fewer epochs). We observe for a large "Python-like" PCFG (Table 1), and a small PCFG with 3 top-level subgrammars (Table 4 in Appendix E.3). That is, losing a few epochs of training to train on subgrammar-only strings seemingly leads the model into a very different, "aligned" subspace of loss-minimized weight-space, although excessive pretraining can eventually reduce gains in final loss (see same Tables).

Why do pretrained models exhibit higher representational similarity across seeds? To probe this, we compare the representational similarity of the top quartile of seeds via cosine similarity of embeddings of three types of sequences: (i) sequences consisting solely of the subgrammar, (ii) sequences with no occurrence of the subgrammar, and (iii) sequences with both the subgrammar and other subsequences. We also compute (iv) the similarity between embedded pairs of a subgrammar sequence and a subgrammar-free sequence. For (i) and (ii), the attention-layers of pretrained models cluster subgrammar sequences (resp. no subgrammar sequences) significantly closer together than directly-trained models. This suggests that substructures learned during pretraining are retained after exposure to the full grammar. Finally, the gap between (iv) and (i), and between (iv) and (ii) is greater

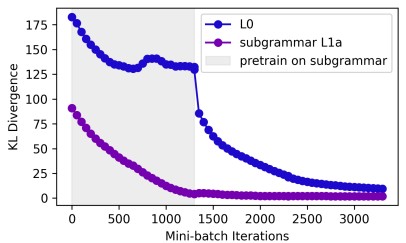
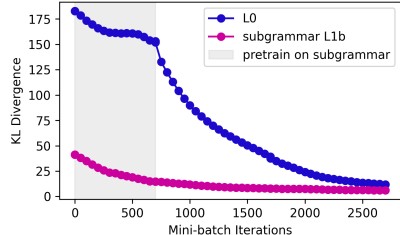
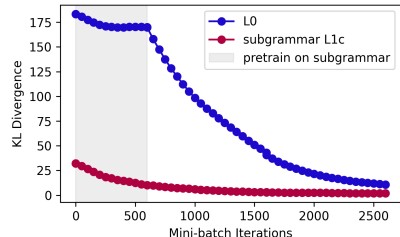

*(a)* Pretraining on a prefix subgrammar      *(b)* Pretraining on an infix subgrammar      *(c)* Pretraining on a suffix subgrammar

*Figure 3.* Examples of pretraining on differently placed subgrammars using `ABC Grammar`.

in pretrained models, suggesting pretrained models are better at internally segregating sequences with and without subgrammar subsequences (Table 3 in Appendix E.3).

Our experiments are not exhaustive, and we leave open the question of *how to train a model to consistently converge to the best optima*, given the rather strong prior of the subgrammar structure of the target CFG. Too little pretraining may not provide a strong enough inductive bias, while too much may over-specialize the model to the subgrammar and hinder transfer. This trade-off mirrors classical insights from curriculum learning, where an optimal "window" of pretraining exposure exists (Bengio et al., 2009; Weinshall et al., 2018).

## 6. Generalization: Do LMs "Know Syntax"?

This paper focuses on CFG *substructure*: introducing subgrammars, establishing the fundamental relationship between subgrammars and language modeling, and initiating the study of how training on substructure affects training dynamics. Another highly related and natural direction is whether models that are loss-minimized on a PCFG truly know, or can generalize, the rules of the CFG. On this front, existing literature in length generalization, theory of transformer expressivity, and LLM performance vs. depth has shown that language models struggle with depth (as opposed to just length). For instance, Delétang et al. (2022) show that transformers (and other architectures) fail to fully generalize on non-regular tasks (in particular PCFGs), LSTMs in principle cannot capture CFGs (Merrill, 2019), and neither can transforms (Hahn, 2020; Bhattamishra et al., 2020), and several papers have shown that transformers fail to generalize to deep / highly-embedded scenarios outside the training distribution (Lakretz et al., 2022). While the above literature is a portal into this interesting subfield, we briefly probe this question with our small transformers trained on an especially simple PCFG: `Nested Parentheses`.

The model achieves very low loss *statistically*. We test generalization to probabilistically unlikely (but grammatically valid) sequences with increasing length in two ways: (i) ex-

tending the context at the same depth of recursion, feeding in $(a)^i$, and (ii) growing sequences through repeatedly applying the recursive rule, resulting in contexts at increasingly deeper depths of recursion, of the form $)^i$. We then compare the model's output logits (its output distribution) against the ground-truth next-token distribution. The next-token distribution is identical for all test contexts, even between cases (i) and (ii).

Figure 4 shows a striking contrast. For case (i), the prediction error remains low throughout, while for case (ii) it grows similarly to an inverse log curve. While the model appears to master the rules of the PCFG at shallow depth, this does not translate into robust handling of deeper recursive dependencies.

We also evaluate the effect of prepending different valid prefixes to the sequence of increasing depth, visualized in Figure 7 Appendix E.4. The results remain largely unchanged – even when using a faulty (non-grammatical) prefix. This suggests that the model's primary difficulty lies in handling the depth of the subsequence it must complete, while it pays relatively little attention to the completed prefix.

Anecdotally, we find similar behavior even in state-of-the-art frontier models. We test GPT-5.1 Instant model on arithmetic expressions generated by a PCFG, presenting two kinds of long expressions: a chain composed of non-deep arithmetic operations, and a single deep arithmetic expression (depth 7)[1]. These tiny probes show that even LLMs, similar to our small LMs, struggle with depth and not length, correctly answering 5/5 non-deep arithmetic expressions but only 2/5 for a deep arithmetic expression. Note that for the not-deep arithmetic expressions (type 1), the LM in fact has to solve more terms than with the deeper recursion, but still solves them correctly.

---

[1]We do not find the same discrepancies for GPT-5.1 Thinking, which solves all of our examples within 3-4 minutes for each expression. The Thinking model may pass arithmetic expressions to a calculator or program, and/or uses an externally prompted or engineered chain-of-thought process; in any case, this departs from language modeling in the strict sense.

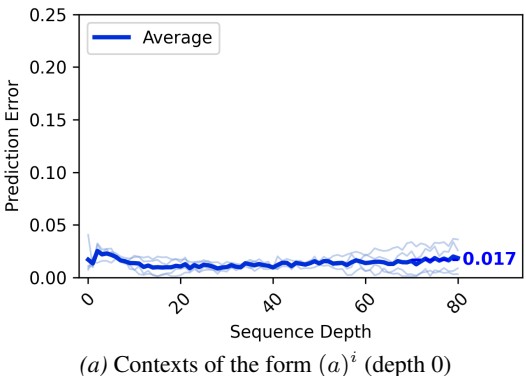

*(a)* Contexts of the form $(a)^i$ (depth 0)

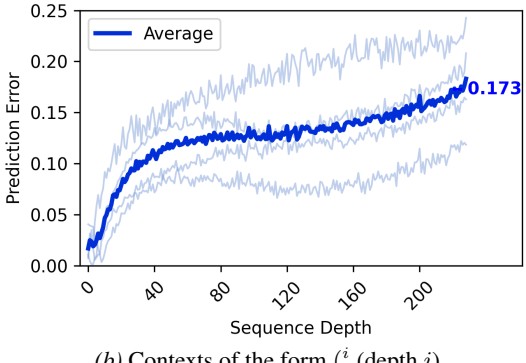

*(b)* Contexts of the form $(^i$ (depth $i$)

*Figure 4.* LM error vs. longer context, with or without recursion

## 7. Discussion and Future Work

Our work initiates the study of language-modeling *dynamics* with respect to the substructure of Context-Free Grammars. On the theoretical side we show that the language-modeling loss decomposes into subgrammar-local contributions. Empirically, we find that small transformers tend to reduce error across many subgrammars in parallel; we offer in a corollary a condition under which gradient descent training has this property, but more generally, there are at least two directions for future work in this subarea: (1) measuring the degree to which concrete architectures satisfy or violate this assumption (including empirically studying whether they hold for architectures other than transformers, e.g. RNNs and LSTMS), and (2) proving parallel learning under weaker conditions.

Our work connects language modeling and substructure of CFGs; these ideas could possibly be extended to other formal classes, in particular ones that extend CFGs such as tree-adjoining grammars, indexed grammars, and most generally context-sensitive grammars. We have also not considered whether more interesting structural decompositions occur in the case of *ambiguity*, i.e. multiple derivation paths of the same token sequences[2].

We also find subgrammar pretraining acts as an inductive bias: in tiny models it can improve performance, and even when it does not, it yields representations *more aligned with the grammar's substructure*; understanding why, or what other benefits this may have is an open question. We have also not studied whether there is a correspondence between specific parts of the model (e.g. certain layers, attention heads) and subgrammar hierarchy.

---

[2]However, Theorems 4.3-4.6 are hold even with ambiguity; if two subgrammars generate the same string, there is still an additive recurrence over the subgrammars. But if one can factor out a shared subgrammar of those subgrammars, one can factor out this subgrammar as a separate term in the top-level recursive decomposition

## 8. Limitations

On the theoretical side, our results thoroughly characterize how language-modeling loss decomposes with respect to subgrammar structure in a variety of cases. However, we do not have a full theory for *when* or *why* gradient-based training yields parallel improvement across subgrammars; towards this, we present a single, somewhat heavy-handed result (Corollary 4.7. Empirically, our experiments use a small set of synthetic PCFGs and scaled-down, decoder-only transformers. Although these grammars are designed to isolate recursion, as well subgrammar sizes and depth, they do not cover the full diversity of CFGs (in particular ambiguity – in all our experiments, CFGs are fully unambiguous; the effect of multiple possible parse-trees presents an interesting direction). Our results of Section 6 leave unresolved whether failures at deep recursion reflect representational limits or optimization barriers: we conjecture that *there exists a setting of the weights* of, say, a 2-layer, 2-head transformer (as in our experiments) that *does* correctly model the PCFG (at least up to some very high bound on depth). This would show that the issue is gradient descent which is not able to find such ideal solutions, analogous to work showing that while neural networks can in principle represent functions like parity, modular counting, or compositional rules, gradient descent often fails to find these solutions without strong inductive bias or curricula (Telgarsky, 2016; Abbe et al., 2024). Finally, we do not compare learning dynamics across other language classes in the Chomsky hierarchy (e.g., regular or mildly context-sensitive languages) under matched conditions, and thus cannot yet disentangle "difficulty of depth" in CFGs from other forms of dependency. Our work also does not explore the question of *grammar induction*, the learning task of determining the CFG underlying the input data.

## Impact Statement

This paper presents work whose goal is to advance the field of machine learning, in particular the theory of language modeling and formal grammars. There are many potential societal consequences of our work, none of which we feel must be specifically highlighted here.

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

| Config | $L$ | $h$ | $d$ | $|\mathcal{V}|$ |
|---|---|---|---|---|
| FourLayer | 4 | 4 | 8 | 100 |
| TwoLayer | 2 | 2 | 20 | 100 |
| TwoLayer_SMALL | 2 | 2 | 6 | 100 |
| TwoLayer_smallVoc | 2 | 2 | 20 | 5 |
| OneLayer | 1 | 1 | 8 | 100 |
| OneLayer_LARGE | 1 | 1 | 32 | 100 |

*Table 2.* Transformer configurations used in our experiments.

## A. Details of our Transformer Architecture

The transformer architectures used in our experiments are scaled-down variants of nanoGPT (Karpathy, 2023). Training proceeds with batches sampled uniformly at random from the dataset. The number of batches per epoch depends on the total size of the training data – this implies that PCFG $G$ which generates longer sequences yield more iterations per epoch. Furthermore, the tokenizers contain only two special tokens: $BOS$ (beginning-of-sequence) and $EOS$ (end-of-sequence). We deliberately omit $UNK$ (unknown) and $PAD$ (padding) tokens, since all tokens are guaranteed to be in the grammar's terminal set $\mathcal{N}$; this ensures the training distribution matches as closely as possible to the grammar distribution.

### A.1. Model Parameter Settings

All models share the same decoder-only Transformer architecture as nanoGPT (Karpathy, 2023). Each model consists of a learned token embedding matrix $E \in \mathbb{R}^{|\mathcal{V}| \times d}$, learned positional embeddings for a fixed context window of 256 tokens, $L$ stacked decoder blocks with multi-head self-attention and a two-layer feed-forward network with hidden size $4d$, followed by a final LayerNorm and a tied output projection $E^\top$. We use GELU activations, dropout rate $p = 0.1$ in the attention, feed-forward, and embedding layers, and LayerNorm with learned scale and bias. Input and output token embeddings are tied, and we exclude the positional embeddings when reporting parameter counts.

We vary the number of layers $L \in \{1, 2, 4\}$, the model dimension $d \in \{6, 8, 20, 32\}$, the number of attention heads $h \in \{1, 2, 4\}$ (with per-head dimension $d/h$), and the vocabulary size $|\mathcal{V}|$, which is determined by the underlying grammar. Table 2 summarizes the configurations used in our experiments.

### A.2. Training and Regularization

All models are trained with the AdamW optimizer using a fixed learning rate of $6 \times 10^{-4}$, $(\beta_1, \beta_2) = (0.9, 0.95)$, and a batch size of 8. We train for a fixed number of epochs.

A central aspect of our training setup is relatively strong weight decay. We use AdamW with an $\ell_2$ penalty $\lambda = 0.1$ applied to all parameters with at least two dimensions (i.e., the token embedding matrix, attention projection matrices, and feed-forward weights), while excluding all bias terms and LayerNorm scale parameters from weight decay. This decoupled weight decay acts as our main form of explicit regularization in addition to dropout ($p = 0.1$ in the attention, feed-forward, and embedding layers) and the small model sizes described in Section A.1. Together, these choices constrain effective capacity and discourage simple memorization of grammar-generated strings.

To further stabilize optimization, we apply gradient norm clipping with a maximum global norm of $1.0$ at every step. We do not use any learning-rate scheduling or warm-up; the learning rate remains constant throughout training. Checkpoints are saved periodically and at the beginning and end of training, allowing us to analyze learning dynamics across epochs.

## B. Details on Centered Kernel Alignment (CKA)

In this section, we provide more detailed information on how we measure representational similarity using *linear* CKA. For each model and each transformer block $\ell$, we register forward hooks on the attention and MLP modules and collect their output activations on the evaluation set. In our experiments, each evaluation sequence is processed separately; if the output of a hooked module has shape $1 \times L \times d_\ell$, we reshape it to $L \times d_\ell$ and concatenate these matrices across all evaluation sequences. Thus, for each model $m$ and submodule / layer $s$, we obtain an activation matrix $H_{m,s} \in \mathbb{R}^{N \times d_s}$, where $N$ is the total number of token positions in the evaluation corpus.

Given two models $m$ and $m'$, we first center activations column-wise,

$$\tilde{H}_{m,s} = H_{m,s} - \frac{1}{N} \mathbf{1}\mathbf{1}^\top H_{m,s},$$

and analogously for $\tilde{H}_{m',s}$. We then compute linear CKA as

$$\text{CKA}(H_{m,s}, H_{m',s}) = \frac{\left\| \tilde{H}_{m,s}^\top \tilde{H}_{m',s} \right\|_F^2}{\left\| \tilde{H}_{m,s}^\top \tilde{H}_{m,s} \right\|_F \left\| \tilde{H}_{m',s}^\top \tilde{H}_{m',s} \right\|_F}.$$

Within each training condition, we compute this quantity between corresponding submodules of all off-diagonal seed pairs. With 30 seeds, this leads to 435 distinct off-diagonal seed pairs. When we report CKA on full-grammar sequences and subgrammar-only sequences, the same procedure is applied separately to the corresponding evaluation subsets.

Linear CKA is well suited to our setting because independently trained models with the same functionality are not expected to match neuron-by-neuron: the same underlying computation may be implemented after a permutation or rotation of the hidden basis. Rather than comparing individual coordinates, CKA compares the geometry induced by activations across token positions. In particular, linear CKA is invariant to isotropic rescaling and orthogonal transformations of the representation space, so high CKA indicates that two models organize the evaluation tokens similarly even when their individual neurons are not directly comparable. We therefore use it as a layerwise summary of cross-seed representational alignment. However, as with any global similarity statistic, CKA does not by itself identify which specific features are shared.

## C. Additional Proofs and Theorems

*Proof of Proposition 3.9.*

$$\mathcal{L}(\theta) = \sum_{x \in \Sigma^*} P(x)(-\log Q_\theta(x)) \tag{1}$$

$$= \sum_{x \in \Sigma^*} P(x)(\log P(x) - \log P(x) - \log Q_\theta(x)) \tag{2}$$

$$= \sum_{x \in \Sigma^*} P(x) \log \frac{P(x)}{Q_\theta(x)} - \sum_{x \in \Sigma^*} P(x) \log P(x) \tag{3}$$

$$= D_{\mathrm{KL}}(P \parallel Q_\theta) - H(P) \tag{4}$$

$$\square$$

*Proof of Theorem 4.1.* The decomposition can be constructed recursively. Given CFG $G = (\Sigma, \mathcal{N}, \mathcal{S}, \mathcal{P}, \mathcal{W})$, the root node of the DAG – initially labeled with only $S$ – represents the entire grammar. If $S$ can generate itself through successive applications of rules of $G$, we add a self-loop from $S$ to itself.

Let $X \subseteq N$ be the subset of non-terminals on the right-hand side of any rule $S \to \alpha$. For each $A \in N$, let $G_A$ be the inner subgrammar generated by taking the closure of $A$ in $\mathcal{P}$ – that is, all the expansions $A \to \alpha$, all expansions of those non-terminals on the right-hand side of those rules, and so on. In the case that the result subgrammar is all of $G$, we can add $A$ as an additional label to the root node. Otherwise, $G_A$ is a proper inner subgrammar, in which case we assign it a node as a child of $S$. Inductively, this procedure is applied to each new subgrammar node (which by construction has strictly fewer non-terminals than its supergrammar). $\square$

*Proof of Theorem 4.3.* Theorems 4.3 and Corollary 4.4 are equivalent, for simplicity we directly prove Corollary 4.4.

Let $A_1, \ldots, A_k$ be the top-level subgrammars of $G$ and suppose $G$ has $r$ rules expanding $S$. Then we can write all rules expanding $S$ as:

$$S \to A_{i(1,1)} \cdots A_{i(1,l_1)}$$

$$\vdots$$

$$S \to A_{i(r,1)} \cdots A_{i(r,l_r)}$$

Suppose each rule in the PCFG occurs with probabilities $p_1, \ldots, p_r$ respectively. As we are directly proving Corollary 4.4, we assume $S$ expands only to non-terminals (by which we will also denote the top-level subgrammars; note that some of these may be $S$ itself if they are not proper subgrammars).

Denoting $P = P_G$ and $Q = Q_\theta$,

$$D_{\mathrm{KL}}(P \parallel Q) = \sum_{s \in \Sigma^*} P(s) \log \frac{P(s)}{Q(s)} \tag{5}$$

$$= \sum_{j=1}^{r} p_j \sum_{a_{j,1}, \ldots, a_{j,l_j}} P(a_{j,1} \cdots a_{j,l_j}) \log \frac{P(a_{j,1} \cdots a_{j,l_j})}{Q(a_{j,1} \cdots a_{j,l_j})} \tag{6}$$

$$= \sum_{j=1}^{r} p_j \sum_{a_{j,1}, \ldots, a_{j,l_j}} P_{A_{i(j,1)}}(a_{j,1}) \cdots P_{A_{i(j,l_j)}}(a_{j,l_j}) \sum_{i=1}^{l_j} \log \frac{P_{A_{i(j,i)}}(a_{j,i})}{Q(a_{j,i}|a_{j,1} \cdots a_{j,i-1})} \tag{7}$$

$$= \Big[ \sum_{j=1}^{r} p_j \sum_{i=1}^{l_j} \sum_{a_{j,1}, \ldots, a_{j,i-1}} P_{A_{i(j,1)}}(a_{j,1}) \cdots P_{A_{i(j,i-1)}}(a_{j,i-1}) \Big] \sum_a P_{A_{i(j,i)}}(a) \log \frac{P_{A_{i(j,i)}}(a)}{Q(a|a_{j,1} \cdots a_{j,i-1})} \tag{8}$$

$$= \sum_{i=1}^{k} \Big[ \sum_s P_G(s|\epsilon) P_G(A|s) \Big] \sum_a P_{A_i}(a) \log \frac{P_{A_i}(a)}{Q(a|s)} \tag{9}$$

$\square$

**Corollary C.1.** *Suppose $G$ has subgrammars $Z_1, \ldots, Z_l$ as irreducible "leaf" subgrammars in its DAG subgrammar decomposition, and all rules evaluate to strings of only non-terminals, or only-terminals. Then*

$$D_{\mathrm{KL}}(P_G \parallel Q_\theta) = \sum_{i=1}^{l} D_{\mathrm{KL}}(P_G \parallel Q_\theta)_{Z_i}$$

*Proof of Theorem 4.6.* Let $G$ be a PCFG with top-level, proper subgrammars $A_1, \ldots, A_k$. Summing over the top-level rules (expansions of $S$), suppose $S$ maps to a rule with $i$ recursive $S$'s with probability $p_i$ ($\sum_{i=0}^{N} p_i = 1$ for some $N < \infty$). Then, $\mathbb{E}[R] = \sum_{i=1}^{N} p_i \cdot i$. Then by Corollary 4.5 (treating both proper subgrammars and recursive $S$ as top-level subgrammars), we have

$$D_{\mathrm{KL}}(P_G \parallel Q_\theta) = \sum_{i=1}^{k} D_{\mathrm{KL}}(P_{A_i} \parallel Q_\theta(A_i)) + \sum_{i=1}^{N} p_i \cdot i D_{\mathrm{KL}}(P_G \parallel Q_\theta) \tag{10}$$

$$= \sum_{i=1}^{k} D_{\mathrm{KL}}(P_{A_i} \parallel Q_\theta(A_i)) + \mathbb{E}[R] D_{\mathrm{KL}}(P_G \parallel Q_\theta) \tag{11}$$

$$\implies D_{\mathrm{KL}}(P_G \parallel Q_\theta) = \frac{\sum_{i=1}^{k} D_{\mathrm{KL}}(P_{A_i} \parallel Q_\theta(A_i))}{1 - \mathbb{E}[R]} \tag{12}$$

$\square$

**Theorem C.2.** *For $G$ with outer subgrammar $A$, let $\bar{A}$ be its complement. The KL-divergence splits as a weighted sum:*

$$D_{\mathrm{KL}}(P_G \parallel Q_\theta) = P_G(A) D_{\mathrm{KL}}(P_A \parallel Q_\theta|_A) + P_G(\bar{A}) D_{\mathrm{KL}}(P_G|_{\bar{A}} \parallel Q_\theta|_{\bar{A}}) + D_{\mathrm{KL}}(P_G^* \parallel Q_\theta^*)$$

*Where $D^*$, for $D \in \{P_G, Q_\theta\}$ is the 2 valued distribution of whether $D$ outputs a string in $A$ or $\bar{A}$, $P_A$ is the language from CFG $A$, and $D|_B$ indicates the marginal distribution of $D$ over strings of $B \in \{A, \bar{A}\}$.*

*Proof.* Writing $P$ for $P_G$ and $Q$ for $Q_\theta$ for legibility,

$$D_{\mathrm{KL}}(P \parallel Q) = \sum_{s \in A} P(s) \log \frac{P(s)}{Q(s)} + \sum_{s \in \bar{A}} P(s) \log \frac{P(s)}{Q(s)} \tag{13}$$

$$= P(A) \sum_{s \in A} P_A(s) [\log P(A) + \log P|_A(s) - \log Q^*(A) - \log Q|_A(s)] \tag{14}$$

$$+ P(\bar{A}) \sum_{s \in \bar{A}} P|_{\bar{A}}(s) [\log P(\bar{A}) + \log P|_{\bar{A}}(s) - \log Q^*(\bar{A}) - \log Q|_{\bar{A}}(s)] \tag{15}$$

$$\tag{16}$$

From which the final decomposition follows quite immediately by rearranging terms. $\square$

## D. Definition of Grammars used for Experiments

In this section we properly introduce the PCFGs used for running the experiments.

### KL decomposition example 1

This grammar concatenates three subgrammars that generate separate spans independently. It is useful for studying settings in which the sequence distribution factorizes cleanly across segments.

$$L1 \rightarrow \texttt{s}L2\_2\ L2\_2\ \texttt{eL2\_2}\ \texttt{s}L2\_1\ L2\_1\ \texttt{eL2\_1}\ \texttt{s}L2\_3\ L2\_3\ \texttt{eL2\_3}\ [1.0]$$

$$L2\_1 \rightarrow NUM\ [0.4]\ |\ L2\_1 \star L2\_1\ [0.15]\ |\ L2\_1 + L2\_1\ [0.15]\ |\ NUM\ NUM\ [0.3]$$

$$L2\_2 \rightarrow \texttt{a}\ L2\_2\ \texttt{b}\ [0.6]\ |\ \texttt{c}\ [0.4]$$

$$L2\_3 \rightarrow \texttt{x}\ L2\_3\ [0.8]\ |\ \texttt{x}\ [0.2]$$

$$NUM \rightarrow \texttt{0}\ [0.2]\ |\ \texttt{1}\ [0.2]\ |\ \texttt{2}\ [0.2]\ |\ \texttt{3}\ [0.2]\ |\ \texttt{4}\ [0.1]\ |\ \texttt{5}\ [0.1]$$

**KL decomposition example 2**

This grammar defines a mixture over the same three subgrammars as in `KL decomposition example 1`, with each sample drawn from exactly one component rather than from their concatenation. It is useful for testing whether a model can represent heterogeneous but mutually exclusive structures.

$$L1 \rightarrow \texttt{s}L2\_1\ L2\_1\ \texttt{eL2\_1}\ [0.3]\ |\ \texttt{s}L2\_2\ L2\_2\ \texttt{eL2\_2}\ [0.3]\ |\ \texttt{s}L2\_3\ L2\_3\ \texttt{eL2\_3}\ [0.4]$$

$$L2\_1 \rightarrow NUM\ [0.4]\ |\ L2\_1 \star L2\_1\ [0.15]\ |\ L2\_1 + L2\_1\ [0.15]\ |\ NUM\ NUM\ [0.3]$$

$$L2\_2 \rightarrow \texttt{a}\ L2\_2\ \texttt{b}\ [0.6]\ |\ \texttt{c}\ [0.4]$$

$$L2\_3 \rightarrow \texttt{x}\ L2\_3\ [0.8]\ |\ \texttt{x}\ [0.2]$$

$$NUM \rightarrow \texttt{0}\ [0.2]\ |\ \texttt{1}\ [0.2]\ |\ \texttt{2}\ [0.2]\ |\ \texttt{3}\ [0.2]\ |\ \texttt{4}\ [0.1]\ |\ \texttt{5}\ [0.1]$$

**Deeper Recursion**

This grammar builds hierarchical structure across multiple explicit levels, with recursive self-concatenation and marked opening and closing symbols at each depth. It is useful for testing deep recursion and long-range boundary matching rather than shallow local regularities.

$$L0 \rightarrow \texttt{s}L1\ L1\ \texttt{eL1}\ [0.7]\ |\ L0\ L0\ [0.3]$$

$$L1 \rightarrow \texttt{s}L2\ L2\ \texttt{eL2a}\ [0.6]\ |\ L1\ L1\ [0.3]\ |\ V\ [0.1]$$

$$L2 \rightarrow \texttt{s}L3\ L3\ \texttt{eL3}\ [0.6]\ |\ L2\ L2\ [0.3]\ |\ V\ [0.1]$$

$$L3 \rightarrow \texttt{s}L4\ L4\ \texttt{eL4}\ [0.6]\ |\ L3\ L3\ [0.3]\ |\ V\ [0.1]$$

$$L4 \rightarrow \texttt{(}\ V\ \texttt{)}\ [0.7]\ |\ V\ [0.3]$$

$$
\begin{aligned}
V \rightarrow\ &\texttt{a}\ [0.04]\ |\ \texttt{b}\ [0.04]\ |\ \texttt{c}\ [0.04]\ |\ \texttt{d}\ [0.04]\ |\ \texttt{e}\ [0.04]\ |\ \texttt{f}\ [0.04]\ |\ \texttt{g}\ [0.04] \\
&\texttt{h}\ [0.04]\ |\ \texttt{i}\ [0.04]\ |\ \texttt{j}\ [0.04]\ |\ \texttt{k}\ [0.04]\ |\ \texttt{l}\ [0.04]\ |\ \texttt{m}\ [0.04]\ |\ \texttt{n}\ [0.04] \\
&\texttt{o}\ [0.04]\ |\ \texttt{p}\ [0.04]\ |\ \texttt{q}\ [0.04]\ |\ \texttt{r}\ [0.04]\ |\ \texttt{s}\ [0.04]\ |\ \texttt{t}\ [0.04]\ |\ \texttt{u}\ [0.04] \\
&\texttt{v}\ [0.04]\ |\ \texttt{w}\ [0.04]\ |\ \texttt{x}\ [0.04]\ |\ \texttt{y}\ [0.04]
\end{aligned}
$$

**Outer Subgrammar Example**

This grammar simplifies the idea of early language learning, such as from children. It places reusable lexical and syntactic components embedded inside a larger sentence frame (i.e. subject – verb – object). This makes it useful for testing how models learn shared structures across different outer contexts.

$$START \rightarrow \texttt{sSUBJ}\ SUBJ\ \texttt{eSUBJ}\ \texttt{sVERB}\ VERB\ \texttt{eVERB}\ \texttt{sOBJ}\ OBJ\ \texttt{eOBJ}\ [1.0]$$

$$SUBJ \rightarrow \textbf{NOUN}\ [0.2]\ |\ \texttt{a}\ NOUN\ [0.4]\ |\ \texttt{the}\ NOUN\ [0.4]$$

$$NOUN \rightarrow \textbf{N}\ [0.7]\ |\ ADJ\ NOUN\ [0.3]$$

$$VERB \rightarrow \mathbf{V} \ [0.3] \ | \ V \ ADV \ [0.7]$$

$$OBJ \rightarrow \mathbf{blank} \ [0.5] \ | \ \texttt{with} \ SUBJ \ [0.5]$$

$$N \rightarrow \texttt{dog}[0.2] \ | \ \texttt{cat}[0.2] \ | \ \texttt{fox}[0.1] \ | \ \texttt{parrot}[0.1] \ | \ \texttt{hamster}[0.1] \ | \ \texttt{turtle}[0.1] \ |$$
$$\texttt{horse}[0.1] \ | \ \texttt{pig}[0.1]$$

$$ADJ \rightarrow \texttt{big}[0.2] \ | \ \texttt{poisonous}[0.2] \ | \ \texttt{cute}[0.2] \ | \ \texttt{lazy}[0.2] \ | \ \texttt{quick}[0.2]$$

$$V \rightarrow \texttt{eats}[0.15] \ | \ \texttt{runs}[0.4] \ | \ \texttt{sleeps}[0.15] \ | \ \texttt{talks}[0.15] \ | \ \texttt{cleans itself}[0.15]$$

$$ADV \rightarrow \texttt{quickly}[0.2] \ | \ \texttt{slowly}[0.3] \ | \ \texttt{happily}[0.3] \ | \ \texttt{excitedly}[0.1] \ | \ \texttt{lazily}[0.1]$$

The rules that are used for the unified subgrammar are highlighted in bold.

## ABC Grammar

This grammar combines several different subgrammars within one sequence, including comparison expressions, balanced dependencies, and simple relative patterns. It is a useful stress test to show how a model can learn multiple compositional regimes simultaneously.

$$L0 \rightarrow \texttt{sL1a} \ L1a \ \texttt{eL1a} \ \texttt{sL1b} \ L1b \ \texttt{eL1b} \ \texttt{sL1c} \ L1c \ \texttt{eL1c} \ [1.0]$$

$$L1a \rightarrow \texttt{sL2a} \ L2a \ \texttt{eL2a} \ L1a \ \texttt{sL2\_2a} \ L2\_2a \ \texttt{eL2\_2a} \ [0.4] \ | \ \texttt{sL2a} \ L2a \ \texttt{eL2a} \ L1a \ [0.2] \ |$$
$$\texttt{action} \ [0.4]$$

$$L1b \rightarrow L1b \ + \ \texttt{sL2b} \ L2b \ \texttt{eL2b} \ [0.25] \ | \ \texttt{sL2b} \ L2b \ \texttt{eL2b} \ [0.75]$$

$$L1c \rightarrow \texttt{xy} \ L1c \ [0.3] \ | \ \texttt{x} \ L1c \ [0.3] \ | \ \texttt{sL2c} \ L2c \ \texttt{eL2c} \ [0.4]$$

$$L2a \rightarrow \texttt{sL3} \ L3 \ \texttt{eL3} \ [0.5] \ | \ \texttt{not} \ L2a \ [0.25] \ | \ L2a \ \texttt{and} \ L2a \ [0.1] \ | \ L2a \ \texttt{or} \ L2a \ [0.15]$$

$$L2\_2a \rightarrow \texttt{a} \ L2\_2a \ [0.8] \ | \ \texttt{a} \ [0.2]$$

$$L2b \rightarrow \texttt{a} \ L2b \ \texttt{b} \ [0.6] \ | \ \texttt{c} \ [0.4]$$

$$L2c \rightarrow \texttt{c} \ L2\_2ac \ [0.7] \ | \ \texttt{c} \ [0.6]$$

$$L3 \rightarrow \texttt{==} \ [0.2] \ | \ \texttt{<=} \ [0.2] \ | \ \texttt{<} \ [0.2] \ | \ \texttt{>=} \ [0.2] \ | \ \texttt{>} \ [0.2]$$

## Nested Parentheses

This is a minimal Dyck-style grammar that generates recursively nested parenthesis with simple leaf symbols. It serves as a test for context-free recursion, stack-like memory, and long-rang matching dependencies.

$$L0 \rightarrow \texttt{(} \ L1 \ \texttt{)} \ [0.7] \ | \ L0 \ L0 \ [0.3]$$

$$L1 \rightarrow \texttt{(} \ L1 \ \texttt{)} \ [0.8] \ | \ \texttt{a} \ [0.2]$$

## PythonPCFG

This is a simplified grammar of Python-like code, combining expressions, assignments, imports, control flow, loops, and function definitions within a hierarchical statement structure. It is useful for testing whether models can learn programming-language style syntax, including nested blocks, indentation-like scope markers, and long-range dependencies between keywords, delimiters, and suites. Additionally, this grammar is more complex and contains a larger vocabulary than the other grammars which is useful to show how the experiments scale with complexity.

$$STMTS \rightarrow stmt \; [0.8] \mid stmt \; STMTS \; [0.2]$$

$$stmt \rightarrow small\_stmt \; \texttt{NL} \; [0.5] \mid compound\_stmt \; [0.5]$$

$$small\_stmt \rightarrow small\_stmt \; small\_stmt \; [0.35] \mid expr\_stmt \; [0.2] \mid return\_stmt \; [0.15] \mid$$
$$import\_stmt \; [0.15] \mid \texttt{PASS} \; [0.05] \mid \texttt{BREAK} \; [0.05] \mid \texttt{CONTINUE} \; [0.05]$$

$$expr\_stmt \rightarrow test \; [0.25] \mid \texttt{NAME EQ} \; test \; [0.5] \mid \texttt{NAME} \; augassign \; test \; [0.25]$$

$$augassign \rightarrow \texttt{+=} \; [0.34] \mid \texttt{-=} \; [0.33] \mid \texttt{*=} \; [0.33]$$

$$return\_stmt \rightarrow \texttt{RETURN} \; [0.3] \mid \texttt{RETURN} \; test \; [0.7]$$

$$import\_stmt \rightarrow \texttt{IMPORT NAME} \; [0.5] \mid \texttt{FROM NAME IMPORT NAME} \; [0.5]$$

$$compound\_stmt \rightarrow compound\_stmt \; compound\_stmt \; [0.2] \mid compound\_stmt\_2 \; [0.8]$$

$$compound\_stmt\_2 \rightarrow if\_stmt \; [0.35] \mid \texttt{FOR NAME IN} \; test : suite \; [0.25] \mid$$
$$\texttt{WHILE} \; test : suite \; [0.2] \mid \texttt{DEF NAME} \; parameters : suite \; [0.2]$$

$$suite \rightarrow simple\_stmt \; [0.9] \mid \texttt{INDENT} \; compound\_stmt\_2 \; \texttt{DEDENT} \; [0.1]$$

$$if\_stmt \rightarrow \texttt{IF} \; test : suite \; [0.35] \mid \texttt{IF} \; test : suite \; \texttt{ELSE} : suite \; [0.35] \mid$$
$$\texttt{IF} \; test : suite \; \texttt{ELIF} \; test : suite \; [0.15] \mid$$
$$\texttt{IF} \; test : suite \; \texttt{ELIF} \; test : suite \; \texttt{ELSE} : suite \; [0.15]$$

$$parameters \rightarrow \texttt{( )} \; [0.4] \mid \texttt{( PARAMS )} \; [0.6]$$

$$PARAMS \rightarrow \texttt{NAME} \; [0.6] \mid \texttt{NAME ,} \; PARAMS \; [0.4]$$

$$test \rightarrow short\_expr \; [0.8] \mid short\_expr \; binop \; short\_expr \; [0.2]$$

$$binop \rightarrow \texttt{+} \; [0.34] \mid \texttt{-} \; [0.33] \mid \texttt{*} \; [0.18] \mid \texttt{/} \; [0.15]$$

$$short\_expr \rightarrow atom\_expr \; [0.85] \mid comparison\_short \; [0.15]$$

$$comparison\_short \rightarrow atom\_expr \; comp\_op \; atom\_expr \; [1.0]$$

$$comp\_op \rightarrow \texttt{EQEQ} \; [0.34] \mid \texttt{NE} \; [0.22] \mid \texttt{LT} \; [0.18] \mid$$
$$\texttt{GT} \; [0.18] \mid \texttt{LE} \; [0.04] \mid \texttt{GE} \; [0.04]$$

$$atom\_expr \rightarrow atom \; [0.7] \mid atom\_expr \; trailer \; [0.3]$$

$$trailer \rightarrow \texttt{( )} \; [0.55] \mid \texttt{(} \; test \; \texttt{)} \; [0.1] \mid \texttt{DOT NAME} \; [0.25] \mid \texttt{(} \; test \; \texttt{)} \; [0.1]$$

$$atom \rightarrow \texttt{NAME} \; [0.48] \mid \texttt{NUMBER} \; [0.27] \mid \texttt{STRING} \; [0.2] \mid$$
$$list\_lit \; [0.03] \mid dict\_lit \; [0.02]$$

$$list\_lit \rightarrow \texttt{( )} \; [0.88] \mid \texttt{(} \; test \; \texttt{)} \; [0.12]$$

$$dict\_lit \rightarrow \texttt{\{ \}} \; [0.9] \mid \texttt{\{} \; test : test \; \texttt{\}} \; [0.1]$$

$$NAME \rightarrow \texttt{x} \; [0.15] \mid \texttt{y} \; [0.15] \mid \texttt{z} \; [0.1] \mid \texttt{n} \; [0.1] \mid \texttt{i} \; [0.1] \mid$$
$$\texttt{j} \; [0.1] \mid \texttt{f} \; [0.1] \mid \texttt{g} \; [0.1] \mid \texttt{val} \; [0.1]$$

$$NUMBER \rightarrow \texttt{0} \; [0.15] \mid \texttt{1} \; [0.15] \mid \texttt{2} \; [0.15] \mid \texttt{3} \; [0.1] \mid \texttt{4} \; [0.1] \mid$$
$$\texttt{5} \; [0.1] \mid \texttt{10} \; [0.1] \mid \texttt{42} \; [0.15]$$

$$STRING \rightarrow \texttt{STR\_A} \; [0.4] \mid \texttt{STR\_B} \; [0.3] \mid \texttt{STR\_HELLO} \; [0.3]$$

# E. Additional Experimental results

## E.1. ChatGPT-5 Instant Arithmetic Stress Test

We generate arithmetic expressions using integers uniformly sampled from 0–9 and the operators {+, -, *, / } are generated. Expression depth is defined as the maximum level of nested parentheses. *Non-deep chains* consist of 50 expressions of depth at most 2, concatenated by addition. *Deep chains* consist of single expressions with recursive nesting up to depth 7. Below are an example each:

**Non-deep arithmetic expression**:
$((4*4)*(1-9))+((6/3)*(5/1))+((2-8)*(8/5))+((5/9)*(7*7))+((7-4)+(8/7))+((9-6)+(1-0))+((0/1)+(9-9))+((4/1)+(0+5))+((6+6)/(2/5))+((4/5)+(0-2))+((3*1)+(5+3))+((1-0)-(7-6))+((2*5)*(5/3))+((6+9)-(6/1))+((1+4)/(6+9))+((9/7)-(6+2))+((6-7)/9)+((4+1)+(7-3))+((5-3)-(1*3))+((5+6)+4)+((5*2)+(0-0))+((6*7)*8)+((5/2)+(4+6))+((5/5)*(9/6))+((4-3)*(8*7))+((7/3)*(9+3))+((7-0)+(5/9))+((6/8)-(2+0))+((0+6)/4)+((9-5)-(3-9))+((0+1)+(9-4))+((7-7)*(1-8))+((7-1)+9)+((4-0)+(0*8))+((6/9)*(2-2))+((5-6)-(8/4))+((3*5)/(4+2))+((3*4)-(5+2))+((7-1)+(8/8))+((4*0)-(9+7))+((3/6)-(4/3))+((0-2)-(1/9))+((0-8)*(8*0))+((0/1)*(2/8))+((9+5)*(8/3))+((1+8)/(4-9))+((0*6)*(2+4))+((5/6)+(2+0))+((2*7)-(2/2))+((8+8)*2)$

Result: $\frac{707449}{1260}$

**Deep arithmetic expression**:
$((((((3+8)+(5-1))-((1-6)+(5+3)))+(((8-2)-(3-8))+((2*9)*(4+5))))*((((1/7)-(6*4))*((7+3)*(6+6)))-(((8*3)*(1+8))+((5-9)+(7/1)))))+(((((8*6)/(5-3))*((8*0)-(8-0)))+(((8-9)+(3-6))-((9/8)/(7*8))))/((((7-4)*(2+2))-((3-5)/(9-2)))/(((6/8)+(5*5))*((4-1)-(8+8))))))-((((((4-0)/(4-8))*((8-0)-(3-1)))+(((7*7)*(4/7))*((7*0)-(0/7))))-((((8*6)/(8+7))-((8/8)+(8/4)))-(((5+5)*(9*8))-((9/2)/(3-9)))))+(((((9-8)+(2*1))-((4+3)/(9-5)))/(((2*2)*(4*3))-((6-6)-(6+9))))*((((8/8)-(3*3))/((8+0)+(9/1)))*(((2*1)*6)*((1+5)/8)))))$

Result: $\frac{892410719}{448320600}$

## E.2. Recursive Decomposition Experiments

Figure 5 shows experimentally the results from Theorem 4.6 on a simple CFG with two rules:

$$S \to x \ \ (p), \quad S \to (S \text{ and } S) \ \ (1 - p)$$

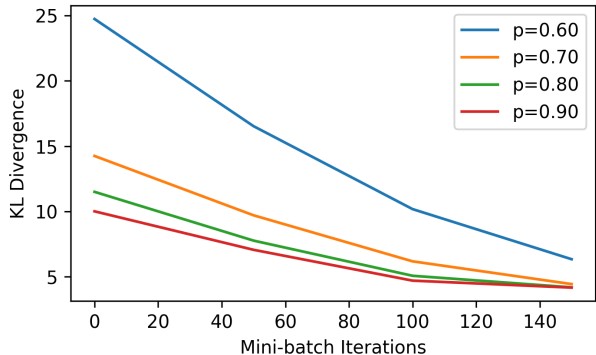

*Figure 5.* Two-layer Transformer showing the impact of the probability of recursion.

## E.3. Pretraining Results

This appendix provides the detailed results referenced in the main text. All experiments compare transformers trained from scratch against those pretrained on a subgrammar before continuing on the full grammar.

Figure 6 illustrates the distribution of KL-divergences across 30 seeds when training directly versus with 10 epochs of subgrammar pretraining. Pretraining consistently shifts the distribution toward lower KL.

Table 3 reports average cosine similarity across attention and MLP layers, on three types of test sequences: (i) sequences consisting solely of subgrammar subsequences, (ii) sequences with no subgrammar subsequences, and (iii) sequences mixing subgrammar and other subsequences.

## E.4. Generalization and Prefix Experiments

This appendix provides the figure referenced in the main text. Figure 7 compares the model's prediction error across three prefix conditions: (a) a valid terminal sequence $(a)(a)(a)(a)(a)(a)$, (b) a valid nested bracket sequence $(((((((a))))))))$, and (c) an invalid non-grammatical sequence $(a)(a)(a))(aa)(a)(a)$. All three conditions yield nearly identical error curves, reaching approximately 0.16-0.17 at depth 200. This demonstrates that the model's performance is primarily determined by the depth of the recursive structure it must complete, with minimal influence from the prefix context or its grammatical validity.

|  | Attention | MLP |
|---|---|---|
| **Sequences with subgrammar only** | | |
| From Scratch | 0.660 | 0.635 |
| With Pretraining | 0.743 | 0.611 |
| *Percentage change (%)* | *+12.6* | *-3.9* |
| **Sequences without subgrammar** | | |
| From Scratch | 0.835 | 0.837 |
| With Pretraining | 0.876 | 0.841 |
| *Percentage change (%)* | *+4.9* | *+0.5* |
| **Sequences with subgrammar** | | |
| From Scratch | 0.726 | 0.501 |
| With Pretraining | 0.687 | 0.543 |
| *Percentage change (%)* | *-5.7* | *+8.4* |

*Table 3.* Average cosine similarity [-1, +1] across attention and MLP layers of a two-layer Transformer when pretraining for 10 epochs. *Percentage change* indicates the relative difference between models trained from scratch and with pretraining.

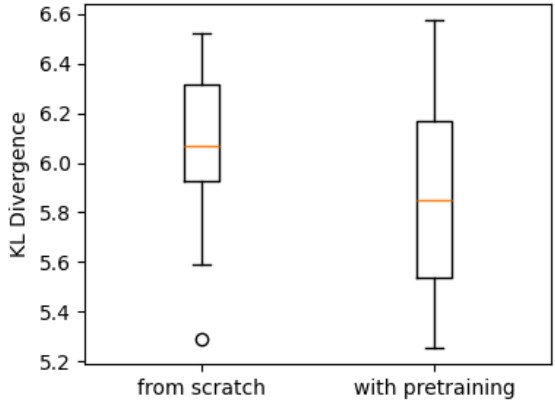

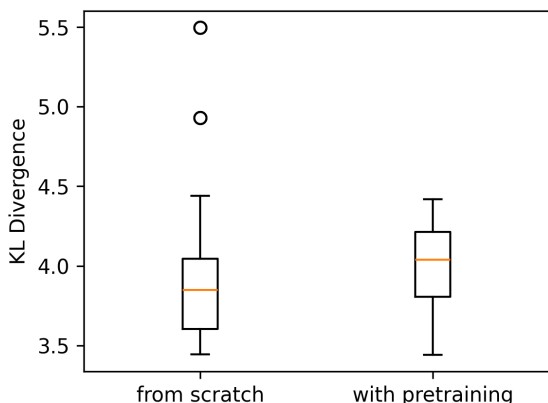

*(a)* Two-layer Transformer

*(b)* Four-layer Transformer

*Figure 6.* Distribution of final KL value of pretraining versus training from scratch

| | Small Two-layer Transformer | | | |
|---|---|---|---|---|
| | Pretraining 2 epochs | | Pretraining 4 epochs | |
| | Attention | MLP | Attention | MLP |
| **Full grammar sequences** | | | | |
| From Scratch | 0.208 | 0.311 | 0.214 | 0.318 |
| With Pretraining | 0.220 | 0.332 | 0.229 | 0.338 |
| *Percentage change (%)* | *+5.8* | *+6.8* | *+7.0* | *+6.3* |
| **Subgrammar sequences** | | | | |
| From Scratch | 0.229 | 0.467 | 0.236 | 0.465 |
| With Pretraining | 0.252 | 0.474 | 0.262 | 0.488 |
| *Percentage change (%)* | *+10.1* | *+1.5* | *+11.0* | *+4.9* |
| Subgrammar pretraining only | 0.248 | 0.543 | 0.593 | 0.944 |

*Table 4.* Average linear CKA similarity (0–1) across attention and MLP layers of different, independently trained transformers when pretraining for 2 vs. 4 epochs. After pretraining, the models were trained for additional 10 epochs. The average was computed over off-diagonal seed pairs (30 seeds; 435 pairs) on the evaluation set. The most significant differences are in teal. This experiment was run using `ABC Grammar`.

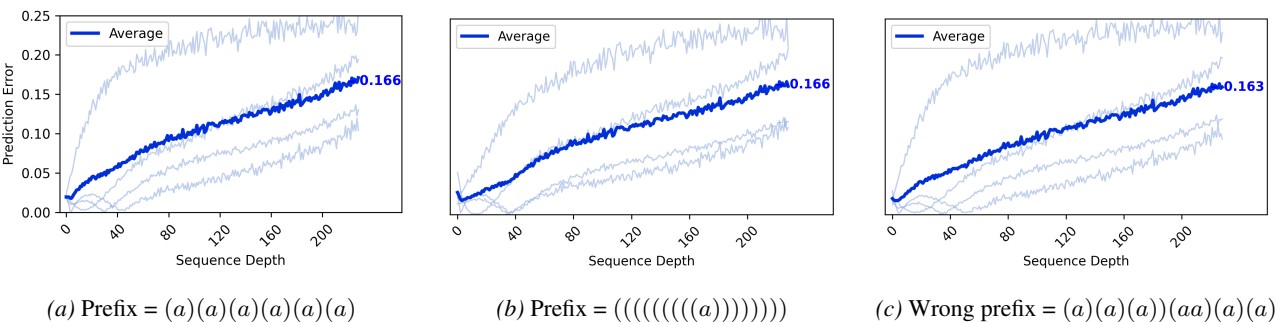

*(a)* Prefix $= (a)(a)(a)(a)(a)(a)$      *(b)* Prefix $= (((((((((a)))))))))$      *(c)* Wrong prefix $= (a)(a)(a))(aa)(a)(a)$

*Figure 7.* Comparison of different prefixes for recursion type 2

