# OpenReview forum: "Unraveling Syntax: Language Modeling and the Substructure of Grammars"
_ICML.cc/2026/Conference — ICML 2026 spotlight_

### Official Review · Reviewer_zFpL · 2026-02-23

**Soundness:** 2
**Presentation:** 2
**Significance:** 2
**Originality:** 4
**Overall Recommendation:** 4
**Confidence:** 2

**Summary:**

This paper studies the learning dynamics of transformer LMs trained on PCFGs. In particular, it defines the notion of a subgrammar - something similar to a subset of a PCFG - and proves that the KL divergence of an LM on a particular PCFG can be decomposed into the sum of its KL divergences on the subgrammars of which that PCFG is composed. One consequence of this is that, if updates to one subgrammar do not harm performance on other subgrammars (along the path of gradient descent) - models will learn all subgrammars at once. The authors then perform experiments showing that this is the case, for small transformers trained on simple PCFGs. They furthermore show that pretraining on subgrammars can improve final loss (for smaller models). They conclude that.

**Compliance With Llm Reviewing Policy:**

Affirmed.

**Final Justification:**

The rebuttal addressed some concerns, and the other reviews indicate that this paper will be of interest to its main target audience.

**Key Questions For Authors:**

Typos:
- derivabl -> derivable
- this implies that ˆθ minimizes θ -> minimizes L(θ)
- straightforrward -> straightforward
- the page supertitle is set to "Submission and Formatting Instructions for ICML 2026" rather than the actual title.
- language modelis ->  language model is
- we then the KL-divergence is C/(2p − 1) for some constant C (missing a word)
- in general, when you reference material (especially Figures) in the appendix, please make it clear that it is in the appendix

**Limitations:**

yes

**Strengths And Weaknesses:**

For context, I work mostly in interpretability; this review mostly concerns this paper's implications with respect to that, and I only understood this paper's proofs at a high level. Feel free to disregard this review in favor of better informed opinions on the paper's formal / theoretical contributions.

**Soundness**: I can't really comment on the soundness of the proofs, though they seem reasonable. The experiments seem a bit weak / thin to me. The authors compare models pretrained on various subgrammars via CKA on their activations, but don't go into detail regarding what they intend to measure using CKA, or why a high CKA should matter at all. What does "similar activations" mean here, and why is this important?

More broadly, the connection between the theoretical and empirical results was not always clear to me; what do the theoretical results have to say about models in practice? The clearest link was in Corollary 4.7; however this relies on the assumption that "that the model and PCFG together obey a kind of “independence”". Is this assumption valid in practice? I guess it may be in the case of small transformers trained on PCFGs, but how does this assumption / corollary translate into the world of LLMs?

Section 6, especially the last paragraph, feels like a hand-wavey non-sequitur. Though I can't recall papers off the top of my head, I think there is past work on this topic (LLM performance vs. syntactic depth) that would support the claims made in the last paragraph better than what's currently there. Re: the failures of small transformers on simple PCFGs, does the transformer expressivity literature have anything to say about this?

**Presentation**: This paper is pretty dense and hard to get through. It asks a lot of readers on the theoretical side, and doesn't always signpost well where its theorems are leading. It also doesn't explain its experiments very thoroughly, frequently omitting key details or explanations of metrics (e.g. CKA). The paper also references many figures in the appendix, without making it explicit that they are located there. At the same time, the paper has a half page of space that it doesn't use; using this space would have made the paper much easier to understand.

In section 5, this paper would do well to cite papers on pre-pretraining, e.g. [Papadimitriou and Jurafsky (2020)](https://aclanthology.org/2020.emnlp-main.554/) or [Hu et al. (2025)](https://aclanthology.org/2025.acl-long.478/); I think Papadimitriou has other relevant work here too.

**Significance**: The question "Why do transformers learn subgrammars all at once, rather than from simple to complex?" (and perhaps also "Why don't we need curriculum learning?") is interesting; however, it's not clear to me that this paper really answers this fully (or proves that the premise is true for transformers beyond small ones). This paper could connect better to the psycholinguistics + language acquisition literature if it wanted to better motivate this question.

**Originality**: This work seems quite original to me, although I don't know the literature that well.

Overall, this paper seems a bit underbaked to me. The presentation could be much improved, and the connection between its theoretical contributions and any real-world results is a bit tenuous, while the theoretical results don't seem interesting enough to justify the entire paper. The experiments run seem a bit cursory as well. I think that a round of significant edits to improve readability, better integrate this paper with the rest of the (non-theoretical) literature, and strengthen the paper's empirical results would do it some good.

---

> ### Author Rebuttal · Authors · 2026-03-31
>
> First of all, thank you for taking the time to read / review our paper and provide helpful feedback. We’ve made several changes that we believe substantially improve our paper. Going in order of your concerns:
>
> “The experiments seem a bit weak / thin to me … What does "similar activations" mean here, and why is this important?”
>
> Good question: in the CKA experiment we train 30 models (30 random initializations) for 10 epochs only on subgrammar, then for an additional 35 epochs on full grammar, 30 models on 20 epochs of subgrammar and 25 on full, and 30 models on 45 epochs of full only. *All models are trained for the same total number of epochs and total sentences*. We find that in the models that spent *less time* on the full language that (1) internal representations of subgrammar strings are much more similar – higher cosine similarity / dot-products of neuron outputs and (2) representations of whole grammar strings are more similar/aligned (this is the most surprising). While we do not have a mechanistic interpretation of this phenomenon (an interesting question for future work), but we *can* say that “losing a few epochs of training to train on subgrammar-only strings seemingly leads the model into a very different, “aligned” subspace of loss-minimized weight-space.” We’ve updated the main text with this discussion and summary of the results, and the appendix with precise definition of CKA analysis.
>
> By the way, to bolster these results, we’ve done the CKA analysis on an *additional*, very different grammar (the previous one was a simplified version of Python, a large grammar with many subgrammars, the new one is a small one with 3 subgrammars). Anonymized link to results: https://drive.google.com/file/d/1buTHgaEUzjW9s1ZCjrUOyjTiGhxrbH0t/view?usp=sharing
>
>
> “More broadly, the connection between the theoretical and empirical results was not always clear to me…”
>
> The paper’s main goal is to introduce the theory of subgrammars, show the fundamental relation between subgrammars and language modeling, validate this empirically with a few small but varied grammars, and then initiate the study of how pretraining with subgrammars (i.e. given the bias of this structure) affects learning dynamics; we show it can improve performance in very constrained models, and more robustly that it leads to the very different “subspace” of weight space (via CKA). The independence of Cor 4.7 appears to be true in our small grammar experiments (and we’ve added one more by another reviewer’s suggestion); we agree whether it holds (or a different version of 4.7) in large / real-world LLMs is an interesting question for future work.
>
>
> “Section 6, especially the last paragraph, feels like a hand-wavey non-sequitur…”
>
> Yes, this is a good point. We intended Section 6 only as a small aside / extra; we have thoroughly rewritten (and shortened) this section, including a discussion of existing literature – please see our response to Reviewer 1 where we respond to  “W3” (control-F for “W3” in our response should work) for the new text of the section (omitting copying it here only due to the character limit, but we can post it in a separate comment too).
>
>
> “This paper is pretty dense and hard to get through…”
>
> We’ll lightly/respectfully push back on “hard to get through” – since this is primarily a theory paper, we wrote it in the style / way of theory papers, i.e. motivating example / derivation of toy version of the main theorem, etc, gradual derivation of more advanced theorem / corollaries. However, to improve we can add signposting to the Introduction summarizing the main theorems on an intuitive level (in “boxed format” around the informal statements of the theorems), and incorporate any other suggestions you have.
>
>
> Also: we’ve improved our explanation of the CKA experiments (see previous description above), fixed appendix referencing throughout the paper with direct hyperlinks to the relevant definitions / figures in the appendix.
>
> “In section 5, this paper would do well to cite papers on pre-pretraining…”
> We have added your references to the paper, thank you!
>
> Finally we’ve fixed all the typos you found (and many more), thank you. We hope you are open to revising your opinion of our paper, as we believe our theoretical results in particular are novel/fundamental and we’re excited to share them with the ICML community.

---

> > ### Author Rebuttal · Reviewer_zFpL · 2026-04-03
> >
> > The authors' rebuttal answers many of my questions. Though this paper could be stronger from an interpretability point of view, it's clear from the other reviews that this paper has something valuable to offer from other perspectives. I've updated my score to indicate this.

---

### Official Review · Reviewer_bWxf · 2026-03-02

**Soundness:** 4
**Presentation:** 3
**Significance:** 3
**Originality:** 3
**Overall Recommendation:** 5
**Confidence:** 2

**Summary:**

The authors study how language models acquire representations of context-free grammars (CFGs). They specifically study *subgrammar* structure of CFGs. They show theoretically that language modeling loss decomposes CFGs into local subgrammars, and find empirically that small transformers actually learn subgrammars in parallel, which is interesting because it is unlike how human children acquire language.

**Compliance With Llm Reviewing Policy:**

Affirmed.

**Final Justification:**

I now better understand the point you are making in Section 5.2, and appreciate the extra experiment. I agree with your point that full-grammar training is already a good baseline, and I don't think you need to run more. I also agree that my point about realistic natural language is outside of the scope of this work, and thus I have increased my score to a full accept.

**Key Questions For Authors:**

1. How exactly is CKA calculated for Table 1? Is it between models from the same initialization?
2. Small nitpick/question: for 4.2, if the KL-divergence is C/(2p-1), across different runs won't C be different because the language model Q_theta will be slightly different each time? I'm sure this doesn't change the results too much, but just checking my understanding.

Typos:
- Abstract: "a limitation that appears even of large language models."
- Line 127 (right column): *straightforrward
- Line 183 (right column): "requires *[that] the strong assumption that the model..."
- Line ~269: modelis
- Line 271: *[we] then the KL-

**Limitations:**

Yes

**Strengths And Weaknesses:**

Strengths:
- Although I am not an expert in theory, the theoretical propositions in the paper appear sound to me, and I appreciate the Figure 1 experiment to show that their math works out as expected.
- The paper feels original; I have not seen many people talking about how the sub-structure of CFGs interacts with language modeling loss, which is intuitive and important.
- The idea that models learn subgrammars in parallel is significant and interesting, especially since it is paired with (toy) empirical results for transformer language models. I would have naively expected a more sequential pattern, so I find this surprising, but also intuitive in retrospect (which is always a nice property of a paper's results).
- The paper is clearly written, with helpful examples and visualizations in Section 4.2.

Weaknesses:
- Why are results in 5.2 surprising? My understanding is that we are measuring CKA across seeds within a given condition: e.g., full grammar CKA between subgrammar pre-trained models. Even if CKA is being measured for full grammar sequences, it makes sense that representations of pre-trained models will be more similar to each other than models with many different random initializations. A baseline here with an equivalent amount of pre-training on something else (possibly arbitrary sequences) might help to show whether this criticism is valid.
- The Section 6 experiments feel a little bit "tacked on," and I think that it was already known that LLMs [struggle with recursion](https://aclanthology.org/2022.coling-1.285/), at least with natural language. (This itself is a little [contested](https://arxiv.org/pdf/2210.15303), and maybe the authors' setup is different, but I think my point still stands that it feels not super related to the rest of the paper).
- It is hard to know whether the results of this paper are useful for modeling realistic natural language (which I assume is the primary motivation here). If this is a primary motivation, then I think the paper needs some extra thought put into how to design an experiment that shows this parallel subgrammar learning happening in more realistic data, perhaps simple stories or child-directed speech for small models. If the authors did this, I would increase my score. If this is not the primary motivation, then I would appreciate more discussion on what these results might be significant for.

---

> ### Author Rebuttal · Authors · 2026-03-31
>
> Thank you for taking the time to read through our paper and give useful feedback, and we have made several changes/additions that we believe substantially improve the paper.
>
> “Why are results in 5.2 surprising? …”
> Good question (we have now clarified/fixed the paper main text): in the CKA experiment *all models are trained for the same total number of epochs and total sentences*. That is, we train 30 models (30 random initializations) for 10 epochs only on subgrammar, then for an additional 35 epochs on full grammar, 30 models on 20 epochs of subgrammar and 25 on full, and 30 models on 45 epochs of full only. We then find that in the models that spent *less time* on the full language that (1) internal representations of subgrammar strings are much more similar/aligned (surprising because these representations are not  “erased” when optimizing for the full grammar) and (2) representations of whole grammar strings are more similar/aligned (this is the most surprising). While we do not have a mechanistic interpretation of this phenomenon (an interesting question for future work), but we *can* say that “losing a few epochs of training to train on subgrammar-only strings seemingly leads the model into a very different, “aligned” subspace of loss-minimized weight-space.”
> To bolster these results / on the suggestions of another reviewer, we’ve done the CKA analysis on an *additional*, very different grammar (the previous one was a simplified version of Python, a large grammar with many subgrammars, the new one is a small one with 3 subrammars). Anonymized link to results: https://drive.google.com/file/d/1buTHgaEUzjW9s1ZCjrUOyjTiGhxrbH0t/view?usp=sharing
>
> We’d be very happy to add a new baseline: do you mean pretraining on random / non-grammatical sequences of tokens, then whole grammar? We can do this, but perhaps the existing base condition of just full-grammar training is similar/better because it *only* presents full grammar strings.  Another thought we had is we could compute CKA in the trained models on *random token sequences*  to show that those representations are *not* aligned differently (with the better explanation of our results, let us know whether/which would be useful).
>
> “The Section 6 experiments feel a little bit "tacked on” …. “
> This is a good point.  We intended Section 6 as a small aside / extra – does a fully trained, loss-minimized model “really know” the subgrammar structure? – and we have simple experiments showing no. We have rewritten and shortened this section -- please see our response to Reviewer 1 where we respond to  “W3” (control-F for “W3” in our response should work) for the new text of the section (omitting copying it here only due to the character limit, but we can post it in a separate comment too).
>
> "It is hard to know whether the results of this paper are useful for modeling realistic natural language … "
> The primary purpose/motivation of our work is to introduce a new theoretical tool/perspective in order to understand language modeling: the compositional substructure of context-free grammars. Our main contributions are theoretical – defining subgrammars, proving the fundamental relations of subgrammars vs. loss/KL, parallel subgrammar learning, and varied (but small scale) demonstration of this empirically,, and an initial foray into whether/how pretraining on substructure affects learning dynamics.
> You are right that this is interesting because natural language is a (mostly, a very complicated) CFG. However we view our work as in line with theoretical/empirical work e.g. studying how neural nets learn polynomials, as a proxy / building block towards understanding how they learn arbitrary functions. Note our theoretical results apply to CFGs of *any* size, and our empirical validations show this unfolding precisely as the theory predicts for various (albeit small) grammars. Finally LLMs are also trained for / highly useful for generating programming languages, and for that one of our grammar examples *is* more realistic – PythonPCFG of the CKA experiment (and we could add a subgrammar loss decomposition plot) is a relatively large simplified version of Python.

---

> > ### Author Rebuttal · Reviewer_bWxf · 2026-04-02
> >
> > Thank you for your response. I now better understand the point you are making in Section 5.2, and appreciate the extra experiment. I agree with your point that full-grammar training is already a good baseline, and I don't think you need to run more. I also agree that my point about realistic natural language is outside of the scope of this work, and thus I have increased my score to a full accept.

---

### Official Review · Reviewer_Wxw6 · 2026-03-12

**Soundness:** 4
**Presentation:** 3
**Significance:** 2
**Originality:** 4
**Overall Recommendation:** 5
**Confidence:** 3

**Summary:**

This paper studies the mathematical relationship between **substructures of context-free grammars (CFGs)** and **language modeling loss**. The authors define two types of **subgrammars**:

* **Inner subgrammars**, corresponding to subtrees of derivation trees.
* **Outer subgrammars**, corresponding to simplified versions of the original CFG.

The central theoretical contribution shows that **language modeling loss (measured as KL divergence)** can be **recursively decomposed along the hierarchy of CFG subgrammars** (Theorems 4.1, 4.3, and 4.6). In particular, **Theorem 4.6** shows that **expected recursion depth amplifies KL divergence**, providing a formal explanation for why recursive structures are difficult to learn.

Empirically, experiments with small transformers show that models **learn all subgrammars in parallel**, that **subgrammar pretraining improves internal representation alignment**, and that **deep recursion—rather than sequence length—is the main bottleneck for generalization**.

**Compliance With Llm Reviewing Policy:**

Affirmed.

**Key Questions For Authors:**

* Can the authors empirically measure the **degree of violation of the context-insensitivity assumption** in trained models? For example, how does conditional subgrammar KL divergence vary across contexts?

* Do **RNN or LSTM models** exhibit the same **parallel learning of subgrammars**, or do they instead learn substructures sequentially?

* How does the subgrammar decomposition behave for **ambiguous CFGs**, where multiple derivation trees correspond to the same string?

* Could similar decomposition ideas extend to **mildly context-sensitive grammars** such as TAG or indexed grammars?

* Is there any correspondence between **subgrammar hierarchy and transformer layer representations**? For example, do particular layers specialize in modeling specific subgrammars?

**Limitations:**

The paper provides a clear and honest limitations discussion. The authors acknowledge several constraints, including the use of **synthetic PCFGs**, **small-scale models**, the restriction to **unambiguous grammars**, and the lack of experiments on other language classes in the Chomsky hierarchy. They also note that it remains unclear whether failures on deep recursion arise from **representational limits or optimization difficulties**.

**Strengths And Weaknesses:**

[Strength]
- The recursive decomposition of KL divergence along the hierarchy of CFG subgrammars provides a novel mathematical connection between **formal language theory and language modeling**. The relationship between **expected recursion and KL divergence amplification** offers a principled explanation for the difficulty of learning recursive structures.

- The empirical results closely match the theoretical predictions. For example, Figure 1 demonstrates that the total KL divergence observed during training decomposes into contributions from individual subgrammars, as predicted by the theory. Such tight theory–experiment alignment is rare and particularly valuable for theoretical work.

- The observation that transformers **learn all subgrammars simultaneously rather than sequentially** provides an interesting insight into neural language learning. This stands in contrast to some theories of human language acquisition and suggests distinctive learning dynamics in neural models.

- The paper is carefully structured, with definitions building logically toward the main theorems. The mathematical exposition is precise and accompanied by intuitive explanations. The running example grammar
( $S \to x \ (p),; S \to (S \text{ and } S)\ (1-p)$ )
helps make the abstract results easier to understand.

- The nested parentheses experiments show a clean contrast: increasing sequence length preserves low error, while increasing **recursion depth** leads to logarithmic error growth. This result helps clarify prior discussions about transformer limitations on formal language tasks.


[Weaknesses]
- All experiments are conducted on grammars with **5–100 terminals** and **small nanoGPT-style models with 1–4 layers**. While this is understandable given the theoretical focus of the work, at least one experiment on either a **larger grammar** or a **simplified PCFG approximation of natural language** would strengthen the practical relevance of the results.
- Several results following Corollary 4.5 rely on the assumption that the model behaves **context-insensitively with respect to subgrammars**. Since autoregressive transformers are fundamentally context-dependent, it would be valuable to measure empirically how closely trained models satisfy this assumption.
- Although the theoretical results apply to autoregressive models in general, experiments are restricted to transformers. A comparison with **RNN or LSTM models** would help determine whether the observed parallel learning behavior is specific to transformers or reflects a broader property of gradient-based learning.
- The benefits of subgrammar pretraining appear substantial for **2-layer models** but diminish for **4-layer models**, suggesting that the practical impact of the approach may decrease with model scale.
- A footnote briefly mentions anecdotal observations from ChatGPT experiments. This informal reference is somewhat inconsistent with the otherwise rigorous tone of the paper and might be better removed or replaced with a controlled experiment.

---

> ### Author Rebuttal · Authors · 2026-03-31
>
> We are thrilled that you liked our paper, and thank you so much for taking the time to read it, review it, and offer suggestions.
>
> "All experiments are conducted on grammars with 5–100 terminals and small nanoGPT-style models with 1–4 layers. While this is understandable given the theoretical focus of the work, at least one experiment on either a larger grammar or a simplified PCFG approximation of natural language would strengthen the practical relevance of the results."
>
> Out of space constraints we didn’t make clear that one of our grammars IS large - namely the one in the CKA experiment – “PythonPCFG” is a simplified (but relatively large) version of Python, for which the CKA analysis took several days to run. This grammar is not a natural language but is an approximation of another large important language that transformers are used for (programming languages). Additionally, we’ve added a new kind of grammar that is structurally very different to our plots for KL decomposition, and also redid the CKA analysis for that same grammar, showing the same intuitive results; training less on the full grammar in order to pretrain on a subgrammar leads to a very different region of low-loss weight-space where not only subgrammar strings are more aligned, but also whole grammar strings. Anonymous links to the new experiments / figures:
> https://drive.google.com/file/d/1iwCtjecsMN4GDvmkcNUZgWoM2r1lwtc-/view?usp=sharing
> https://drive.google.com/file/d/1buTHgaEUzjW9s1ZCjrUOyjTiGhxrbH0t/view?usp=sharing
> Note that we’ve also added the definition and an intuitive description of PythonCFG to the paper to emphasize that is larger / an approximation of a programming language.
>
> [q1] Can the authors empirically measure the degree of violation of the context-insensitivity assumption in trained models?
>
> In our small experiments, we found this to be the case – whether we computed a model’s loss/KL on a subgrammar averaged over different prefixes or with a fixed, simple prefix, the decomposition results looked identical. This is because our models are quite overparameterized with respect to the languages, fulfilling Cor 4.7. But a minor point of the small “extra” section 6 is that, with rare / out of distribution contexts, this breaks down entirely (the LM’s output is arbitrarily far from the subgrammar distribution).
>
> [q2] Do RNN or LSTM models exhibit the same parallel learning of subgrammars, or do they instead learn substructures sequentially?
>
> Our theoretical results of course hold for ANY language model and the “overparametrization” result of Cor 4.7 applies for any gradient-descent based learning algorithm– but whether it holds in practice or not is a great question for future work! We have added the idea to the discussion.
>
> “A footnote briefly mentions anecdotal observations…”
> This is a good point, yes we are happy to remove it if you think it improves the paper (we thought it might be more exciting / motivating if we had even an anecdotal example from state of the art models)
>
> “How does the subgrammar decomposition behave for ambiguous CFGs…”
> This is a good question. Theorems 4.3-4.6 are "independent" of ambiguity– if two subgrammars generate the same string, there is still an additive recurrence over these grammars; if once could factor out a shared grammar (i.e. equivalent DAG with a subgrammar with two in-edges) one could factor out a shared subgrammar as a separate term in the recursive decomposition. We have added a footnote mentioning this. However, there may be more interesting theory relating to ambiguity – a good topic for future work.
>
> “Could similar decomposition ideas extend to mildly context-sensitive grammars such as TAG or indexed grammars?”
> Well… you guessed our main idea for a follow-up project and paper, specifically indexed grammars which are a better model for natural language, and in which we suspect elegant recursive formulae to hold for loss (but we have not shown this yet – in fact, the notion of sub-index-grammar needs to be defined properly).
>
> “Is there any correspondence between subgrammar hierarchy and transformer layer representations?”
> You guessed our other main question for future work – this one we have done some preliminary work suggesting that yes (e.g. that certain attention heads are more selective to some subgrammars).
> We have added these ideas (with credit to a helpful reviewer who also had these ideas) to our discussion section.

---

> > ### Author Rebuttal · Reviewer_Wxw6 · 2026-04-02
> >
> > Thank you for the authors’ detailed response. I appreciate the clarification regarding the scale of PythonPCFG, the additional experiments on structurally different novel grammars (including KL decomposition and CKA analysis), and the expanded discussion of ambiguous CFGs and mildly context-sensitive grammar extensions. These additions improve the overall completeness of the paper.
> >
> > That said, I still find a few points insufficiently addressed. In particular, the rebuttal did not discuss the reduced effect of subgrammar pretraining at the larger model scale (4-layer), and the degree of violation of the context-insensitivity assumption was discussed only qualitatively, without quantitative evidence.
> >
> > Overall, I will maintain my current score.

---

### Official Review · Reviewer_mgQT · 2026-03-12

**Soundness:** 3
**Presentation:** 3
**Significance:** 4
**Originality:** 4
**Overall Recommendation:** 5
**Confidence:** 4

**Summary:**

In this paper, the authors propose a new perspective for thinking about the learning of grammars by language models. This perspective is based around subgrammars (intuitively, components of a larger grammar). The authors formalize the notion of a subgrammar and then prove several theorems about how the loss on the overall grammar can be decomposed into losses over subgrammars under certain conditions. They then show several empirical findings relating to this broad framing: (i) overall loss is indeed the sum over subgrammar losses; (ii) language models learn subgrammars in parallel rather than sequentially; (iii) pretraining on subgrammars provides a helpful inductive bias; and (iv) models fail at generalizing recursive grammars to high depths of recursion.

**Compliance With Llm Reviewing Policy:**

Affirmed.

**Final Justification:**

I have increased my score from 4 to 5 because the rebuttal addressed the presentational concerns that I had.

Since the soundness, originality, significance, and clarity were all high in my assessment after the rebuttal, the weighting of them is not important, as any weighting would yield a high score.

**Key Questions For Authors:**

Q1. Would it be feasible to extend the key empirical results to more grammars to show the robustness of the conclusions? What I particularly have in mind would be extending results shown in Figure 1 and Figure 2 (i.e., since those cover 2 grammars already, maybe add a third grammar as well?), and then extending Table 1 to 2 more gramamrs? [This question would be likely to increase my score]

Q2: In Section 5.1, as far as I can tell, the motivation includes prefix subgrammars but the results only include infix subgrammars and suffix subgrammars. How do prefix subgrammars perform? [This question is nice to have but is unlikely to change my score]

Q3: Would it be feasible to move a bit more information and results into the main paper? I think it would be particularly helpful to move Figure 5 into the main paper, and also to add in the main paper a brief description of the grammars in Appendix C (including a pointer to Appendix C for the full grammars). [This question would be likely to increase my score]

**Limitations:**

yes

**Strengths And Weaknesses:**

Strengths:
S1 (significance, originality): The paper provides a useful framing of grammar learning through the lens of subgrammars. This framing - and its formalization - might be useful to the community by paving the way for interesting future work that builds on this concept.

S2 (soundness): The paper includes proofs of several key concepts relating to subgrammars, providing a solid theoretical foundation for the perspective that the authors argue for.

S3 (soundness, significance): The paper includes a range of empirical results supporting the theoretical framing. I particularly liked the findings showing how the loss decomposes into subgrammar losses. These results cover a range of *types* of results, which is useful for showing the breadth of usefulness of the paper’s framing (i.e., results that validate the theory; results on improving model performance via inductive biases; and results on evaluating generalization).

S4 (presentation): The paper is overall clear and well-written, laying out the concepts in a logical way that connects to prior literature in both formal language theory and neural networks and also building logically across the paper’s empirical and theoretical contributions. (There are some exceptions to this strength, which I note below under weaknesses).

Weaknesses:
W1 (presentation): Many key details were in the appendices. In general I don’t object to this, except that in some cases the main paper did not say that the details were in the appendices, so I thought these details were just missing until I looked through the appendices - what I specifically have in mind is the definitions of the grammars (Appendix C). This could be fixed by making sure that the main paper points to the appendices where relevant.

W2 (soundness): Most of the empirical results are based on just one or two grammars. Given that the paper is about CFGs in general, it would strengthen the paper to show that each key finding holds across multiple grammars (e.g., three meaningfully different grammars for each key finding). I recognize that this would be more work, but hopefully not too much more work given the simplicity of the setting?

W3 (originality): The finding about models failing to generalize to greater depths relates to well-studied limitations in length generalization in neural networks learning formal languages. This is not a big problem, since the paper has other contributions that are very original, but it would be good to at least point to the literature on length generalization. Some relevant papers are:
Deletang et al.: “Neural networks and the chomsky hierarchy”
Merrill: “Sequential Neural Networks as Automata”
Yao: “Self-Attention Networks Can Process Bounded Hierarchical Languages”

W4 (presentation): The paper has a fairly large number of typos. These generally did not impede understanding, so it’s not a significant problem, but it would be good to proofread the paper thoroughly. Here are the typos that I noticed that did pose some challenges for understanding:
Line 112, right column: should “minimizes \theta” be “minimizes L(\theta)”?
Definition 4.2: Should “P(s|\epsilon)” be “P_G(s|\epsilon)”?
Line 192, left column: should “strings only non-terminals” be “strings of only non-terminals”?

W5 (soundness): Section 5.1 is motivated by contrasting prefix subgrammars to infix and suffix subgrammars, but results are only shown for infix and suffix subgrammars.

Overall, I think this is a very strong paper that is weakened in its current version by some presentational issues and by some limited scope of the experiments. I think these issues are very addressable, and I would consider raising my score if they are addressed.

---

> ### Author Rebuttal · Authors · 2026-03-31
>
> First of all, thank you for your thorough review and feedback! We have carefully went through and incorporated your suggestions which we believe has improved our paper substantially:
> W1:
> This is a good catch and “nostra culpa” for not referencing the appendix better. We have fully changed this and have added a short referencing section at the end of the introduction and direct (verbal) hyperlinks to appendix subsections whenever relevant. E.g. references to proofs of theorems are now also hyperlinked to the proofs in appendix, same for figures and tables, ...
>
> Q1 / W2:
> This is one of the main pieces of feedback we have implemented to improve our paper. First, we added KL (loss) decomposition for an additional grammar (the “ABC grammar”). We’ve created a fully anonymous google drive to link to the new figure: https://drive.google.com/file/d/1iwCtjecsMN4GDvmkcNUZgWoM2r1lwtc-/view?usp=sharing.
> With this we now have empirical / visual demonstrations of Theorems 4.3 / Corollary 4.5 for 3 varied kinds of grammars: Figure 1 for a grammar with many top-level subgrammars, Figure 2 with deep grammars nested one within another, the new Figure of a grammar with subgrammars all of which occur with *varying* probabilities (and show KL sub-losses scaled by these probabilities add to total KL loss) (note separately we also have Figure 2b and Figure 6 for other theorems).
>
> We also did the CKA experiment with an additional grammar– the ABC grammar and its subgrammar (L1a) https://drive.google.com/file/d/1buTHgaEUzjW9s1ZCjrUOyjTiGhxrbH0t/view?usp=sharing (this is really 2 new experiments, since we do 2 epochs of pretraining, and 4 epochs of pretraining). We again find that training on subgrammar sequences again “leads gradient descent” into a very different loss valley of high alignment. All models are trained on the same total number of sequences (for the new grammar and previous one), which means models-without-pretraining are trained over more sequences of the full language; nevertheless, activations are more similar in the models which spent epochs on subgrammar-only strings for *both* subgrammar strings *as well* whole-grammar sequences. We’ve updated the main text with the previous sentences to make clearer the main takeaway.  Note, the ABC grammar is very different than the one we had; the previous grammar PythonPCFG, is a simplified version of Python but as such is relatively large (the CKA experiment takes multiple days on it), with many kinds of subgrammars and complex structure (and hence why we train on more epochs); on the other hand the ABC grammar is a small grammar with 3 top-level subgrammars.
>
>
> W3: This is a good point.  We intended Section 6 as a small aside / extra – does a fully trained, loss-minimized model “really know” the subgrammar structure? – and we have simple experiments showing no. But we’ve rewritten this section (and shortened slightly, to allow moving other content from appendix into paper):
> “This paper focuses on CFG \emph{substructure}: introducing subgrammars, establishing the fundamental relationship between subgrammars and language modeling, and initiating the study of how training on substructure affects training dynamics. Another highly related and natural direction is whether models that are loss-minimized on a PCFG truly know, or can generalize, the rules of the CFG.
> On this front, existing literature in length generalization, theory of transformer expressivity, and LLM performance vs. depth has shown that language models struggle with depth (as opposed to just length).  For instance, (Deletang et al) show that transformers (and other architectures) fail to fully generalize on non-regular tasks (in particular PCFGs), LSTMs in principle cannot capture CFGs (Merrill), and neither can transforms (Hahn 2020, Bhattamishra et al 2020), and several papers have shown that transformers fail to generalize to deep / highly-embedded scenarios outside the training distribution (Shizhuo Zhang et al, Lakretz et al).
> While the above literature is a portal into this interesting subfield, we briefly probe this question with our small transformers and PCFGs. … (the rest of the text is shortened summary of Figures 4 and 8)”
>
>
> W4: All typos fixed, including the specific ones you point out and several more!
> Q2 / W5: Good catch. We probably showed the results from the infix subgrammar and suffix subgrammars because they are more surprising (that they’re not forgotten), but we will add back in the plot for the prefix subgrammar.
> Q3: We’ve moved Figure 5 to the main text. Next, we have added intuitive descriptions of the grammars to the main text in the figures. Figure 1 legend now says “S-> L2 L2_2 L2_3, that is, a grammar with 3 top-level subgrammars all occurring with probability 100%”, Figure 2 says “Deeper Recursion: a grammar containing a subgrammar containing a subgrammar … 4 times”, and the new KL-decomposition says “a grammar with inner subgrammars, each occurring with probability < 100%”.

---

> > ### Author Rebuttal · Reviewer_mgQT · 2026-04-03
> >
> > Thank you for the helpful rebuttal! I have increased my score.

---

### Decision · Program_Chairs · 2026-04-30

**Decision:**

Accept (spotlight)

**Comment:**

This paper studies the interactions between different and subsets of a PCFG during training. It theoretically proves and empirically validates in small settings a claim that models will tend to learn PCFG subgrammars—assuming that they are compatible with each other and not in competition—simultaneously.

This is an interesting finding, although I expect at more realistic scale, we would see changes in both the primary empirical finding and the related empirical results using similarity analysis. Nonetheless, the paper provides novel theoretical tools for reasoning about complex structured sequential data.